# AFORA: ACTIVATION-AWARE FACTORIZATION WITH OPTIMAL RANK ALLOCATION FOR TRAINING-FREE LLM COMPRESSION

## ABSTRACT

Large language models are challenging to deploy because of their extreme size and compute demands. In this work, we propose *AFORA* (Activation-aware Factorization with Optimal Rank Allocation for Training-free LLM Compression), a simple and hardware-friendly framework that directly reduces the number of parameters through low-rank factorization of weight matrices. *AFORA* consists of two core components: (1) **Activation-aware Weight Factorization (AWF)**, a closed-form low-rank approximation that accounts for the input activation distribution to preserve task-relevant directions and ensure numerical stability; and (2) **Optimal Rank Allocation (ORA)**, a global rank allocation strategy that assigns heterogeneous ranks across layers to minimize activation distortion under a given budget. Evaluations across multiple large-scale language models show that our framework consistently outperforms existing approaches at the same compression ratios, while additionally reducing model size, saving memory, and decreasing computation with hardware-friendly layer dimensions. It also requires only a short runtime to perform compression, and offers a principled mathematical interpretation. These results demonstrate that activation-aware, globally optimized low-rank compression offers a practical and theoretically grounded path to efficient LLM deployment.

## 1 INTRODUCTION

Large language models (LLMs) have achieved remarkable success (Achiam et al., 2023; Brown et al., 2020; Touvron et al., 2023), but their rapidly growing parameter counts pose severe challenges for memory, compute, and deployment efficiency. To mitigate this, many efforts have attempted to reduce the number of parameters through various compression techniques. Among existing approaches, **quantization** reduces precision to save memory. However, even after quantization, model execution typically requires dequantization, meaning that inference-time acceleration gains are minimal while the primary benefit lies in storage reduction. To fully exploit quantization for runtime acceleration, one must rely on specialized hardware kernels, which are not always available or portable across devices (Frantar et al., 2022; Xiao et al., 2023; Lin et al., 2024; Yao et al., 2022). **Pruning** reduces parameter counts by removing weights or blocks, yet width pruning produces sparse matrices that rarely translate to proportional speedups and still require full memory loads. Depth pruning has been proposed as a more structured variant that removes entire blocks without changing the overall architecture. However, naively discarding less important blocks can overly simplify the model and compromise robustness (Han et al., 2015; Hoefler et al., 2021; Song et al., 2024; Dettmers et al., 2023).

These observations highlight the need for an alternative that reduces parameter counts in a hardware-friendly manner, avoids architectural modifications, and still preserves fine-grained modeling capacity. In other words, instead of simply eliminating less important components, we argue for mathematically principled methods that replace computations with efficient low-rank structures, thereby reducing operations while approximating the original behavior to minimize performance loss.

**We propose *AFORA* (Activation-aware Factorization with Optimal Rank Allocation for Training-free LLM Compression)**, a simple yet principled framework that directly reduces pa-

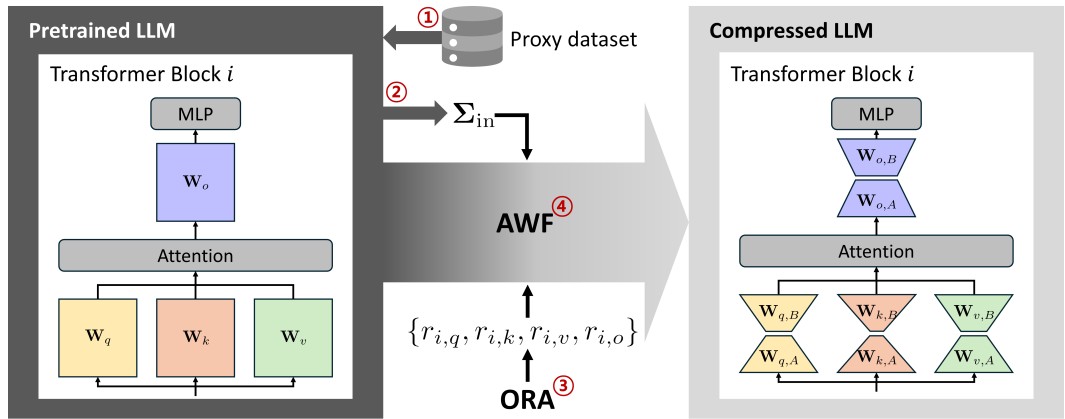

Figure 1: Overview of **AFORA**

rameters through low-rank factorization of less important weight matrices. Unlike quantization and pruning, *AFORA* is training-free, hardware-efficient, and supported by a theoretically grounded analysis that accounts for activation geometry. In practice, it also keeps the compression stage lightweight and leads to inference-time reductions that are consistent with the nominal FLOPs savings.

Our method is built upon two core developments: (i) **Activation-aware Weight Factorization (AWF)**, a closed-form low-rank approximation guided by the input activation distribution, preserving task-relevant directions and ensuring numerical stability. (ii) **Optimal Rank Allocation (ORA)**, a global allocation strategy that assigns rank across layers optimally under a given budget.

**Our contributions are threefold:**

- We propose a new weight factorization method, referred to as **AWF**. We provide a rigorous justification from both a statistical perspective and an optimization viewpoint, showing that it minimizes reconstruction loss in a principled manner.
- We develop the optimal rank allocation strategy that assigns heterogeneous ranks across layers. We theoretically prove that it yields the mathematically optimal rank assignment under a global compression budget.
- We demonstrate the effectiveness of the proposed **AFORA** framework through extensive experiments on `LLaMA-2-7B`, showing that at a compression ratio of 0.15, **AFORA** reduces perplexity on WikiText-2 by 10.27% compared to the best existing method. It also reduces compression time by up to 9.60× relative to the same baseline, achieves an inference speedup of 1.21×, and matches the zero-shot accuracy of the dense model.

Together, these results show that activation-aware, globally optimized low-rank compression provides a practical and theoretically grounded path to efficient LLM deployment.

## 2 BACKGROUND AND RELATED WORK

### 2.1 LOW-RANK APPROXIMATION AS A COMPRESSION TOOL

Low-rank approximation has long been used to reduce the complexity of large linear operators. Given a weight matrix $\boldsymbol{W} \in \mathbb{R}^{d_{\text{out}} \times d_{\text{in}}}$, it can be factorized as $\boldsymbol{W} \approx \boldsymbol{BA}$ with $\boldsymbol{B} \in \mathbb{R}^{d_{\text{out}} \times r}$ and $\boldsymbol{A} \in \mathbb{R}^{r \times d_{\text{in}}}$, where $r \ll \min(d_{\text{out}}, d_{\text{in}})$. This reduces both parameter count and FLOPs approximately as $d_{\text{out}} d_{\text{in}} \rightarrow r(d_{\text{out}} + d_{\text{in}})$. Because Transformers are dominated by large linear layers (attention projections and feed-forward blocks) Vaswani et al. (2017), low-rank approximation offers a hardware-friendly pathway to reduce model size and compute (FLOPs). The most common variant is truncated singular value decomposition (SVD), which selects the top-$r$ singular components of $\boldsymbol{W}$ under the Frobenius norm. By the **Eckart–Young–Mirsky theorem**, this truncated

SVD is the unique optimal rank-$r$ approximation in weight space (Eckart & Young, 1936; Mirsky, 1960). A formal statement and proof sketch are provided in Appendix A.

**Limitations of naive SVD.** Truncated SVD is simple and training-free but *data-agnostic*: it minimizes the reconstruction error of $\boldsymbol{W}$, not the induced error on outputs. Directions that seem unimportant in $\boldsymbol{W}$ may be heavily amplified by input activations, leading to large output errors.

## 2.2 ACTIVATION-AWARE REDUCTION: VIEWPOINT AND PRIOR ATTEMPTS

**Activation-aware viewpoint.** Compression quality should be measured in the output space, under the input activation distribution. This motivates a whitened operator formulation (formalized later in Sec. 3). Concretely, the objective is

$$\min \ \left\| \boldsymbol{W}\mathbf{x} - \boldsymbol{B}\boldsymbol{A}\mathbf{x} \right\|_F^2, \tag{1}$$

which emphasizes minimizing the output error induced by the low-rank factorization.

**Prior heuristics.** Several works attempt to improve upon naive SVD by incorporating task-related statistics. **FWSVD** (Hsu et al., 2022) modifies the spectrum using Fisher information so that directions aligned with high curvature are preserved. **ASVD** (Yuan et al., 2023) instead rescales components according to the empirical activation distribution. Beyond such modifications of the spectrum, **LoRAP** (Li et al., 2024) highlights the heterogeneity across submodules: attention projections can tolerate low-rank designs better, while MLP layers are more sensitive to compression. **SVD-LLM** (Wang et al., 2024) further shows that even activation-aware variants still suffer significant performance loss, which must be compensated by parameter update. These findings suggest that while heuristics can partially mitigate the shortcomings of naive SVD, they are still largely heuristic in nature and lack a unified theoretical foundation. This indicates room for deeper mathematical analysis of activation-aware low-rank approximations—a direction we pursue in this work.

## 2.3 OTHER COMPRESSION PARADIGMS (TRAINING-FREE, HARDWARE-FRIENDLY)

We now contrast our approach with other paradigms that, like ours, can be applied **without training**. Our focus is on post-training compression methods that can reduce parameters or computations without requiring additional fine-tuning.

**Quantization.** Quantization reduces model size by lowering numerical precision and is especially attractive in post-training settings. Different designs vary in whether quantization is applied uniformly or in a group-wise manner, and whether activations are calibrated to control variance. At moderate bit-widths (e.g., 8–4 bits), quantization can achieve strong compression with minimal accuracy loss. However, quantized inference typically requires dequantization before matrix multiplication, which means that real efficiency gains often rely on specialized kernels. This introduces hardware-dependence and limits portability. For ultra-low precision (e.g., 2–3 bits), the theoretical gains often depend on specialized kernels, and practical speedups are not consistently realized across hardware platforms (Frantar et al., 2022; Xiao et al., 2023; Lin et al., 2024; Yao et al., 2022; Dettmers et al., 2023; Egiazarian et al., 2025).

**Pruning.** Pruning has been an important line of research for compressing LLMs, as it can reduce parameter redundancy and lower computational cost. There are two main forms: **unstructured pruning**, which removes individual weights and can reach very high sparsity levels (often above 90%), but struggles to deliver real acceleration due to irregular memory access and the need to fully load parameters into memory; and **structured pruning**, which removes predefined groups (e.g., rows, columns, or 2:4 patterns) to form hardware-friendly patterns. While structured pruning is more practical for deployment, its actual speedup is often much smaller than the theoretical pruning ratio, especially at small batch sizes, and preserving linguistic performance typically requires substantial retraining. Recent studies also combine pruning with low-rank approximations, suggesting that pruning can complement other compression techniques. Overall, pruning remains a valuable tool, but its limitations in robustness and hardware efficiency highlight the need for simpler and more principled alternatives beyond pruning (Sanh et al., 2020; Michel et al., 2019; Lagunas et al., 2021; Song et al., 2024; Wee et al., 2025).

**Low-rank factorization.** Low-rank approximation factorizes large weight matrices into smaller components, reducing both parameter counts and multiply–accumulate operations while retaining standard dense matrix multiplications. Its use in model compression dates back to early work on convolutional networks, and its effectiveness in language models is supported by evidence that their intrinsic dimensionality is much smaller than the parameter space. Classic truncated SVD provides a closed-form solution but is weight-space and data-agnostic. Recent refinements introduce activation statistics or focus on submodule specialization, alleviating some limitations but still lacking a unified theoretical foundation or global optimization strategy. In contrast, our method explicitly formulates compression in whitened activation space, yielding a mathematically justified loss-aware factorization, and combines this with optimal global rank allocation to distribute budgets across layers in a principled way (Hsu et al., 2022; Yuan et al., 2023; Li et al., 2024; Wang et al., 2024).

## 3 ACTIVATION-AWARE WEIGHT FACTORIZATION

### 3.1 FORMULATION FROM ACTIVATION DISTRIBUTION

We begin with the activation-aware objective:

$$\min_{\text{rank}(BA) \leq r} \mathbb{E}_x \left[ \|Wx - BAx\|_F^2 \right], \tag{2}$$

which measures compression error directly in the output space under the input distribution. As shown in Appendix B, this is equivalent to

$$\min_{\text{rank}(BA) \leq r} \left\| W\Sigma_{\text{in}}^{1/2} - BA\Sigma_{\text{in}}^{1/2} \right\|_F^2, \tag{3}$$

where $\Sigma_{\text{in}} = \mathbb{E}[xx^\top]$ is the input covariance. Letting $\Sigma_{\text{in}} = Q\Lambda Q^\top$ with $\Lambda \succeq 0$, we compute $\Sigma_{\text{in}}^{\pm 1/2} = Q\Lambda^{\pm 1/2}Q^\top$ and define the whitened operator

$$T = W\Sigma_{\text{in}}^{1/2} = U \operatorname{diag}(s) V^\top. \tag{4}$$

By the Eckart–Young–Mirsky theorem, the optimal rank-$r$ approximation is the truncated SVD

$$T_r^\star = U_r S_r V_r^\top, \qquad B^\star A^\star = U_r S_r V_r^\top \Sigma_{\text{in}}^{-1/2}. \tag{5}$$

We call this solution the **Activation-aware Weight Factorization (AWF)**.

### 3.2 LOSS-BASED DERIVATION

**Second-order expansion of the loss.** Consider a weight matrix $\boldsymbol{W}$ and perturbation $\Delta\boldsymbol{W}$. Expanding the training loss around $\boldsymbol{W}$ yields

$$\Delta\mathcal{L} = \mathcal{L}(\boldsymbol{W} + \Delta\boldsymbol{W}) - \mathcal{L}(\boldsymbol{W})$$
$$\approx \langle \nabla_{\boldsymbol{W}}\mathcal{L}(\boldsymbol{W}), \Delta\boldsymbol{W} \rangle + \tfrac{1}{2} \operatorname{vec}(\Delta\boldsymbol{W})^\top \boldsymbol{H}_{\boldsymbol{W}} \operatorname{vec}(\Delta\boldsymbol{W}), \tag{6}$$

where $\boldsymbol{H}_{\boldsymbol{W}} = \nabla_{\boldsymbol{W}}^2 \mathcal{L}(\boldsymbol{W})$. For a converged (near-stationary) pretrained model, the *first-order term* is typically small, so the loss increase is well approximated by

$$\Delta\mathcal{L} \approx \tfrac{1}{2} \operatorname{vec}(\Delta\boldsymbol{W})^\top \boldsymbol{H}_{\boldsymbol{W}} \operatorname{vec}(\Delta\boldsymbol{W}). \tag{7}$$

**K-FAC approximation.** For a linear layer $h = \boldsymbol{W}x$, the Kronecker-factored curvature (K-FAC) approximation (Martens & Grosse, 2015; Grosse & Martens, 2016; Botev et al., 2017) gives

$$\boldsymbol{H}_{\boldsymbol{W}} \approx \Sigma_{\text{out}} \otimes \Sigma_{\text{in}}, \qquad \Sigma_{\text{in}} = \mathbb{E}[xx^\top], \quad \Sigma_{\text{out}} = \mathbb{E}[gg^\top], \ g = \nabla_h \ell(h). \tag{8}$$

Using the identity $\operatorname{vec}(MXN) = (N^\top \otimes M)\operatorname{vec}(X)$, the quadratic form in equation 7 becomes

$$\Delta\mathcal{L} \approx \tfrac{1}{2} \left\| \Sigma_{\text{out}}^{1/2} \Delta\boldsymbol{W} \Sigma_{\text{in}}^{1/2} \right\|_F^2. \tag{9}$$

**Practical choice $\Sigma_{\text{out}} = \boldsymbol{I}$.** Estimating $\Sigma_{\text{out}}$ requires backpropagated gradients $g$, which is costly at LLM scale. In practice we set $\Sigma_{\text{out}} = \boldsymbol{I}$ to keep the method training-free, which yields

$$\Delta\mathcal{L} \approx \tfrac{1}{2} \left\| \Delta\boldsymbol{W} \, \Sigma_{\text{in}}^{1/2} \right\|_F^2, \qquad \text{i.e.,} \quad \min_{\text{rank}(\boldsymbol{BA}) \leq r} \left\| \boldsymbol{W}\Sigma_{\text{in}}^{1/2} - \boldsymbol{BA} \, \Sigma_{\text{in}}^{1/2} \right\|_F^2, \tag{10}$$

which is the activation-aware covariance objective used throughout.

### 3.3 INFORMATION-BALANCED FACTORIZATION

While AWF gives the optimal approximation in expectation, its factors $B$ and $A$ can in practice exhibit scale imbalance across directions. Such imbalance may amplify numerical sensitivity and reduce stability, especially when singular values or activation statistics are highly anisotropic. To mitigate this issue, we introduce **Information-Balanced Factorization**, which balances per-direction scales. We formulate the problem as

$$\min_{\alpha_i>0} \ \max_i \left\{ \|B(\alpha)_{:,i}\|_2, \ \|A(\alpha)_{i,:}\|_2 \cdot \|(\Sigma_{\text{in}}^{1/2}V_r)_{:,i}\|_2 \right\}, \tag{11}$$

where $B(\alpha) = U_r\text{diag}(\alpha)$ and $A(\alpha) = \text{diag}(s_i/\alpha_i)V_r^\top\Sigma_{\text{in}}^{-1/2}$. This criterion enforces balanced scaling across the left factor, the right factor, and the whitened input directions, preventing any single component from dominating.

The closed-form solution is

$$\alpha_i^\star \ = \ \sqrt{s_i \cdot \|(\Sigma_{\text{in}}^{1/2}V_r)_{:,i}\|_2}, \qquad i = 1,\ldots,r, \tag{12}$$

which equalizes scales across factors by taking the geometric mean of the singular value and the whitened input norm, thereby improving conditioning. See Appendix C for a detailed derivation.

## 4 ORA: OPTIMAL RANK ALLOCATION

### 4.1 WHY GLOBAL RANK ALLOCATION?

AWF (Sec. 3) specifies the best rank-$r$ approximation *within* a layer once $r$ is fixed. In practice, however, we must decide *how much* rank each layer should be allocated under a single memory or FLOPs budget. Naive strategies such as uniform ranks or hand-crafted schedules ignore two key observations from AWF: (i) whitened spectra differ widely across layers, inducing different marginal utilities (see Appendix E); (ii) the per-rank cost depends on the layer's shape. This motivates a global allocator that explicitly trades off *value* against *cost* across layers, akin to classical knapsack formulations (Kellerer et al., 2004).

### 4.2 GLOBAL OPTIMIZATION OBJECTIVE

Let targets be indexed by $m \in \{1,\ldots,M\}$ with weights $\boldsymbol{W}_m \in \mathbb{R}^{d_{\text{out},m} \times d_{\text{in},m}}$. With AWF (using $\boldsymbol{\Sigma}_{\text{out}} = \boldsymbol{I}$), define the whitened operator

$$\boldsymbol{T}_m \ = \ \boldsymbol{W}_m \boldsymbol{\Sigma}_{\text{in},m}^{1/2} \ = \ \boldsymbol{U}_m \text{diag}(\boldsymbol{s}_m) \boldsymbol{V}_m^\top, \qquad \boldsymbol{s}_m = (s_{m,1} \geq s_{m,2} \geq \cdots). \tag{13}$$

Retaining the top $r_m$ directions yields utility $F_m(r_m) = \sum_{i=1}^{r_m} v_{m,i}$ with per-direction utilities $v_{m,i} = \frac{1}{2}s_{m,i}^2$. The per-rank cost is $c_m = d_{\text{in},m} + d_{\text{out},m}$, corresponding to parameter counts; FLOPs scale in the same order but differ by a constant factor depending on sequence length. Given a global budget $\mathcal{B}$, we solve

$$\max_{\{r_m \in \mathbb{Z}_{\geq 0}\}} \ \sum_{m=1}^{M} F_m(r_m) \qquad \text{s.t.} \quad \sum_{m=1}^{M} c_m r_m \leq \mathcal{B}. \tag{14}$$

### 4.3 LAYER-WISE NORMALIZATION

To remove spurious scale differences across layers while preserving within-layer order, we normalize utilities by the leading singular value:

$$\tilde{v}_{m,i} \ = \ \frac{v_{m,i}}{v_{m,1}} \ = \ \frac{s_{m,i}^2}{s_{m,1}^2}, \qquad \tilde{F}_m(r) = \sum_{i=1}^{r} \tilde{v}_{m,i}. \tag{15}$$

This preserves the relative importance of directions within a layer while placing all layers on a comparable scale, similar in spirit to normalization in PCA and related factor models (Jolliffe & Cadima, 2016).

## 4.4 CONVEX RELAXATION AND WATER-FILLING

Relax equation 14 to $r_m \in \mathbb{R}_{\geq 0}$ with Lagrange multiplier $\lambda \geq 0$. KKT stationarity yields a *global density cutoff*:

$$\tilde{v}_{m,r_m^\star}/c_m \geq \lambda^\star \quad \text{and} \quad \tilde{v}_{m,r_m^\star+1}/c_m < \lambda^\star, \quad \forall m, \tag{16}$$

i.e., keep all directions whose density (utility per cost) exceeds $\lambda^\star$. In discrete form this is the familiar water-filling rule (Tse & Viswanath, 2005):

$$\text{keep } (m,i) \text{ iff } \tilde{v}_{m,i}/c_m \geq \lambda^\star, \qquad \sum_m c_m r_m(\lambda^\star) = \mathcal{B}. \tag{17}$$

Because $\tilde{v}_{m,i}$ is nonincreasing in $i$, selections are per-layer prefixes. Under this prefix property, a standard exchange argument shows that the relaxation admits an integral optimum; see Appendix D for details.

## 4.5 RANK FLOORS AND CONSTRAINED GLOBAL ALLOCATION

We allow per–block floors $r_m^{\min}$ for blocks $m \in \{1, \ldots, M\}$. Let $k_m := \text{rank}(\boldsymbol{W}_m)$, per–rank cost $c_m > 0$, global budget $\mathcal{B}$, values $v_{m,i}$ (e.g., $v_{m,i} = \frac{1}{2}s_{m,i}^2$), and densities $\rho_{m,i} := v_{m,i}/c_m$.

**Feasibility.** The floor–constrained problem

$$\max_{\{r_m \in \mathbb{Z}_{\geq 0}\}} \sum_{m=1}^{M} \sum_{i=1}^{r_m} v_{m,i} \quad \text{s.t.} \quad r_m \geq r_m^{\min} \ (\forall m), \ \sum_{m=1}^{M} c_m r_m \leq \mathcal{B}$$

is feasible iff

$$0 \leq r_m^{\min} \leq k_m \quad (\forall m), \qquad \sum_{m=1}^{M} c_m r_m^{\min} \leq \mathcal{B}. \tag{18}$$

**Allocation when feasible.** Pre–assign floors and water–fill the remainder.

$$\mathcal{B}' \leftarrow \mathcal{B} - \sum_{m=1}^{M} c_m r_m^{\min}, \tag{19}$$

$$r_m^\star = r_m^{\min} + r_m^+(\lambda^\star), \tag{20}$$

$$r_m^+(\lambda) := \max\left\{r \leq k_m - r_m^{\min} \ : \ \frac{v_{m, r_m^{\min}+r}}{c_m} \geq \lambda\right\}, \tag{21}$$

where $\lambda^\star$ is chosen (by bisection) so that $\sum_m c_m r_m^+(\lambda^\star) = \mathcal{B}'$. Equivalently, keep all remaining directions with $\rho_{m,i} \geq \lambda^\star$. Algorithm 1 summarizes the full procedure. The final tie-fill step resolves ties that occur at the density cutoff: if multiple layers have the same marginal density at $\lambda^\star$, we increment the one with the largest next density by one rank, repeating this until the budget is fully consumed.

**Result and connection to AWF.** Applying AWF with the ORA ranks $\{r_m^\star\}$ yields $\boldsymbol{B}_m^\star \boldsymbol{A}_m^\star = \arg\min_{\text{rank} \leq r_m^\star} \|\boldsymbol{W}_m \boldsymbol{\Sigma}_{\text{in},m}^{1/2} - \tilde{\boldsymbol{W}}_m \boldsymbol{\Sigma}_{\text{in},m}^{1/2}\|_F^2$. By construction (additivity of the whitened quadratic loss), ORA maximizes $\sum_m \tilde{F}_m(r_m)$ under $\mathcal{B}$, giving the smallest second-order loss increase among all rank assignments with the same cost. A full proof of optimality under the prefix property is provided in Appendix D.

## 5 EXPERIMENTS

We evaluate our framework *AFORA* (AWF + ORA) under strict **training-free** conditions. All results are obtained without gradient updates, using only a small calibration stream to estimate input covariances. We compare against standard training-free baselines and conduct ablations to isolate the effects of each component.

---

**Algorithm 1** ORA: Water-filling with Normalized Utilities and Floors

---

**Require:** Targets $\{\boldsymbol{W}_m\}_{m=1}^M$, costs $\{c_m\}$, budget $\mathcal{B}$, floors $\{f_m\}$
**Ensure:** Assigned ranks $\{r_m^\star\}$

1: **Per-layer spectra:** For each $m$, estimate $\boldsymbol{\Sigma}_{\text{in},m}$ on calibration data; form $\boldsymbol{T}_m = \boldsymbol{W}_m \boldsymbol{\Sigma}_{\text{in},m}^{1/2}$; compute singular values $\boldsymbol{s}_m = (s_{m,1}, s_{m,2}, \dots)$.
2: **Normalized utilities:** $\tilde{v}_{m,i} \leftarrow s_{m,i}^2/s_{m,1}^2$ for all $i$.
3: **Floors & feasibility:** $B_{\min} \leftarrow \sum_m c_m f_m$; if $\mathcal{B} < B_{\min}$, abort (infeasible).
4: **Initialize bisection:** $\lambda_{\min} \leftarrow 0$, $\lambda_{\max} \leftarrow \max_{m,i} \tilde{v}_{m,i}/c_m$.
5: **while** $\lambda_{\max} - \lambda_{\min} > \epsilon$ **do**
6: $\quad \lambda \leftarrow (\lambda_{\min} + \lambda_{\max})/2$.
7: $\quad r_m(\lambda) \leftarrow f_m + \#\{i > f_m : \tilde{v}_{m,i}/c_m \geq \lambda\}$ for all $m$.
8: $\quad$ **if** $\sum_m c_m r_m(\lambda) > \mathcal{B}$ **then**
9: $\quad\quad \lambda_{\min} \leftarrow \lambda$
10: $\quad$ **else**
11: $\quad\quad \lambda_{\max} \leftarrow \lambda$
12: $\quad$ **end if**
13: **end while**
14: **Tie-fill:** Set $r_m^\star \leftarrow r_m(\lambda^\star)$. While $\sum_m c_m r_m^\star < \mathcal{B}$, increment the $m$.
15: **return** $\{r_m^\star\}$.

---

## 5.1 EXPERIMENTAL SETUP

**Model.** We compress LLaMA-2-7B, LLaMA-3-8B (Touvron et al., 2023), and QWEN-3-8B (Bai et al., 2023), each evaluated in the bfloat16 configuration for consistency across methods. Each model consists of 32 decoder layers with multi-head attention and feed-forward blocks. Our compression is applied to the attention projections (query, key, value, and output matrices). Unless noted otherwise, the same compression ratio is applied across the full model using our global allocator.

**Datasets and metrics.** For language modeling we report perplexity on WikiText-2 (Merity et al., 2016) and Penn Treebank (PTB) (Marcinkiewicz, 1994). For zero-shot reasoning we evaluate accuracy on ARC-Easy (Clark et al., 2018), HellaSwag (Zellers et al., 2019), PIQA (Bisk et al., 2020), WinoGrande (Sakaguchi et al., 2020), CB (De Marneffe et al., 2019), OpenbookQA (Mihaylov et al., 2018), and GSM8K (Cobbe et al., 2021). Evaluation follows the LM Evaluation Harness protocol (Gao et al., 2024). Perplexity is measured with standard left-to-right generation, while reasoning accuracy is reported as multiple-choice accuracy without any in-context examples.

**Compression ratio and budgets.** We report compression ratios as the fraction of parameters removed from the dense model. Since only the attention projection matrices are compressed, ranks are determined using our ORA algorithm, which constructs the whitened spectra of each projection matrix and identifies the retained directions via a density cutoff obtained through bisection search. Costs are measured in parameter count and therefore scale directly with the sum of retained ranks across layers. For example, in the case of LLaMA-2, a 0.15 compression ratio corresponds to removing about 47.07% of the attention weights, which proportionally reduces the average ranks of the projection matrices.

**Computation cost.** The computational cost of estimating $\Sigma_{\text{in}}$ is proportional to a single forward pass over the calibration set. This step scales linearly with the number of calibration sequences. The subsequent SVD is performed once per linear projection layer, matching the cost structure of other SVD-based compression methods.

**Hardware and software environment.** All experiments are run on a server equipped with 4 NVIDIA RTX 6000 Ada Generation GPUs (48GB memory each), CUDA 12.4, and driver version 550.144.03. Models are evaluated in mixed-precision (bfloat16) with PyTorch 2.3 and Hugging-Face Transformers 4.44. Multi-GPU evaluation is handled with the Accelerate library. Reported perplexities and accuracies are averaged over three independent runs to reduce variance.

## 5.2 MAIN RESULTS

**Perplexity under compression.** Table 1 reports perplexity at 15% compression. At this budget, our method exhibits the least performance degradation, maintaining results close to the dense model while all SVD-based baselines degrade substantially. Compared with ASVD, our approach achieves the same compression ratio with $9.6\times$ faster compression time, highlighting its practical efficiency in large-scale settings. All compression times were measured on a single RTX 6000 Ada GPU under the experimental setup described in Section 5.1. The large gap in compression time stems from how ranks are assigned: ASVD determines ranks by exhaustively testing multiple compression ratios for every layer and measuring the resulting perplexity drop, which is computationally expensive. In contrast, AFORA performs a single global optimization step—constructing layerwise utilities and identifying the density cutoff through an $O(\log N)$ bisection search—allowing it to match ASVD's performance while reducing compression time by an order of magnitude.

Table 1: Language modeling performance at 15% compression ratio on `LLaMA-2-7B`.

| Method | Compression Ratio in MHA ↑ | PPL(WikiText-2)↓ | PPL(PTB)↓ | Compression time↓ |
|---|---|---|---|---|
| Dense | 0 | 10.46 | 128.01 | |
| SVD | 0.47 | 851.48 | 2512.70 | 435 *sec* |
| FWSVD | 0.47 | 940.68 | 1896.48 | 322 *sec* |
| ASVD | | 13.15 | 147.69 | 14604 *sec* |
| SVD-LLM | | 15.12 | 174.37 | 963 *sec* |
| SLEB (prune) | | 13.58 | **141.54** | |
| **AWF** | 0.47 | 12.88± 0.10 | 162.26± 2.37 | 1127 *sec* |
| *AFORA* | 0.47 | **11.80**± 0.06 | 148.70± 2.88 | 1522 *sec* |
| Repeat for compression ratio 5% and 10% in Appendix F.1. | | | | |

**Zero-shot reasoning.** Table 2 shows that AFORA preserves accuracy across multiple reasoning benchmarks. Across most tasks, our method consistently ranks first or second in accuracy, confirming that it preserves generalization capability. At the same time, we observe a small but consistent pattern: AFORA tends to perform better on truth-judgment tasks such as CB, whereas ASVD sometimes holds an edge on more specialized or multi-step reasoning benchmarks like OpenbookQA and GSM8K.

Table 2: Zero-shot evaluation at 15% compression ratio on `LLaMA-2-7B`.

| Method | ARC-E | HellaSwag | PIQA | WinoGrande | CB | OpenbookQA | GSM8K | Avg. |
|---|---|---|---|---|---|---|---|---|
| Dense | 0.71 | 0.70 | 0.79 | 0.68 | 0.48 | 0.28 | 0.13 | 0.54 |
| SVD | 0.29 | 0.35 | 0.49 | 0.50 | 0.41 | 0.12 | 0.00 | 0.31 |
| FWSVD | 0.37 | 0.56 | 0.67 | 0.62 | 0.41 | 0.22 | 0.00 | 0.41 |
| ASVD | **0.62** | **0.65** | 0.71 | 0.69 | 0.45 | **0.28** | **0.03** | **0.49** |
| SVD-LLM | 0.57 | 0.60 | 0.66 | **0.71** | 0.41 | 0.22 | 0.02 | 0.46 |
| SLEB (depth) | 0.54 | 0.53 | 0.67 | 0.58 | 0.41 | 0.22 | 0.00 | 0.42 |
| **AWF** | 0.52 ± 0.05 | 0.59 ± 0.03 | 0.68 ± 0.01 | 0.70 ± 0.02 | 0.38 ± 0.04 | 0.17 ± 0.01 | 0.01 ± 0.01 | 0.44 ± 0.00 |
| *AFORA* | 0.61 ± 0.01 | 0.64 ± 0.02 | **0.74** ± 0.01 | 0.66 ± 0.01 | **0.52** ± 0.05 | 0.23 ± 0.02 | 0.01 ± 0.00 | **0.49** ± 0.00 |
| Repeat for compression ratio 5% and 10% in Appendix F.2. | | | | | | | | |

Table 3 shows that **AWF** is the dominant contributor, giving the largest perplexity reduction compared to SVD. **ORA** degrades performance when applied on top of SVD, indicating that naive rank allocation cannot compensate for poor factorization. This is because the optimization objective used in ORA is derived from the activation-aware formulation of AWF. As a result, ORA becomes meaningful only when paired with AWF, where it successfully refines rank assignments and yields further

improvements. Table 4 further confirms that replacing dense linear matrices with low-rank counterparts provides consistent inference-time gains in terms of wall-clock latency, and the measured acceleration closely follows the theoretical reduction in FLOPs for the multi-head attention modules. For clarity, the FLOP counts follow the standard cost of the attention projections. In particular, the dense model requires $O(4Ld^2)$ operations for the $q, k, v, o$ projections, whereas the low-rank variant uses $O(2LdR)$ for rank-$R$ factors. All reported values in Table 4 use sequence length $L = 1024$ and batch size 1. Importantly, this trend highlights that our method achieves computation savings that scale in proportion to, or even beyond, the nominal compression ratio, whereas many alternative approaches only offer sublinear reductions in FLOPs and wall-clock time relative to their compression rates (Li et al., 2023). This proportionality between compression and actual computational gain underscores a key strength of our approach over competing methods. Additional ablations highlighting the specific role of ORA—both its behavior on top of ASVD and its layerwise rank patterns—are provided in Appendix F.2 and Appendix J.

Table 3: Ablation study on `LLaMA-2-7B`.

| Ratio | Metric | SVD | AWF | SVD + ORA | **AFORA** |
|---|---|---|---|---|---|
| 5% | PPL ↓ | 94.67 | 862.40 | 10.97 ± 0.05 | **10.95** ± 0.04 |
| 10% | PPL ↓ | 310.16 | 1624.27 | 11.58 ± 0.06 | **11.36** ± 0.05 |
| 15% | PPL ↓ | 851.48 | 3409.46 | 12.88 ± 0.10 | **11.80** ± 0.06 |

Table 4: Wall-clock speedup.

| Compression Ratio | FLOPs$_{MHA}$ | Speedup |
|---|---|---|
| dense | 1.00× ↓ | 1.00× ↑ |
| 0.05 | 0.84× | 1.09× |
| 0.10 | 0.69× | 1.15× |
| 0.15 | 0.53× | 1.21× |

### 5.3 SENSITIVITY ANALYSES

We report two supplementary analyses related to calibration size and cross-architecture behavior.

**Effect of calibration set size.** All experimental results use 200 single-sequence batches (batch size = 1, sequence length = 1024) to estimate $\Sigma_{in}$. To assess sensitivity to this choice, we vary the number of calibration batches while keeping all other settings fixed. Figure 2 shows that performance stabilizes once the batch count exceeds roughly 100. This suggests that our default setting (200 batches) is conservative but provides a reliably stable estimate of $\Sigma_{in}$.

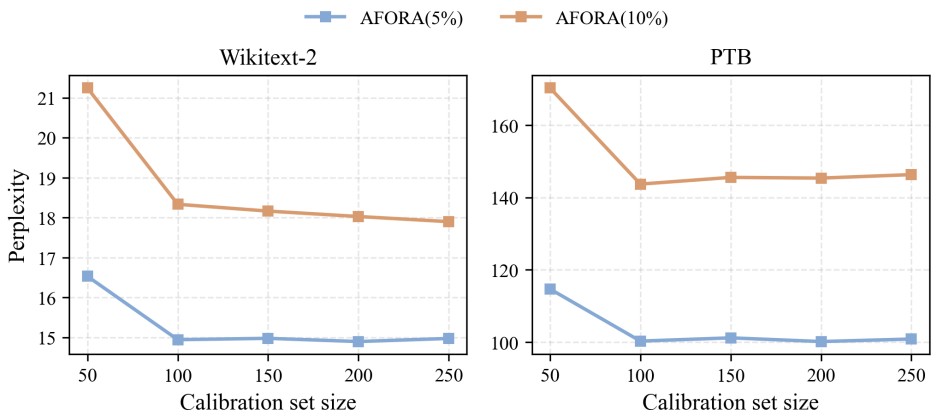

Figure 2: The effect of calibration set size.

**Cross-model generality.** We additionally apply AFORA to `LLaMA-3-8B` and `Qwen-3-8B` under the same training-free pipeline and comparable compression ratios. Both models show the same qualitative trend as LLaMA-2: AFORA consistently outperforms weight-space SVD and achieves stable perplexity under a fixed budget. All detailed results—including full perplexity tables, zero-shot accuracy—are provided in Appendix G, H.

## 6 FUTURE WORK

Our study opens several directions for further improvement and broader application of activation-aware, optimally allocated low-rank compression.

### 6.1 TOWARDS FULLY LOSS-AWARE FORMULATIONS

For tractability, we assumed $\Sigma_{\text{out}} = I$ in the quadratic loss approximation. A natural extension is to incorporate nontrivial output covariances, either estimated from calibration data or approximated via Fisher information. This would yield a more refined objective

$$\|\Sigma_{\text{out}}^{1/2}(W - BA)\Sigma_{\text{in}}^{1/2}\|_F^2 \tag{22}$$

capturing gradient sensitivities in addition to input statistics.

### 6.2 INTEGRATION WITH OTHER COMPRESSION PARADIGMS

While AFORA is training-free and hardware-friendly, it can be combined with pruning and quantization. Structured pruning can remove inactive blocks first, after which AFORA allocates ranks only to the remaining projections, avoiding unnecessary factorization. For quantization, the ordering matters: applying quantization before AFORA introduces noise into the weight matrices and distorts the whitened operator, leading to suboptimal utilities and rank decisions. In contrast, applying quantization after AFORA is often more stable, as the information-balanced factorization reduces outlier directions in the factors $(B, A)$ and can therefore lower quantization error.

### 6.3 ALTERNATIVE ALLOCATION STRATEGIES

Beyond the water-filling allocator, we also plan to evaluate the **incremental allocation** algorithm as an alternative (Fox, 1966). Unlike water-filling, which admits an efficient closed-form solution, the incremental allocation proceeds in discrete increments and is known to incur higher computational complexity. By comparing the two approaches, we aim to characterize the complexity–performance tradeoff and assess whether the incremental scheme offers any practical benefits in the context of rank allocation.

## 7 CONCLUSION

We introduced **AFORA**, a training-free and activation-aware framework for LLM compression. Our approach is built on two key components:

1. **Activation-aware reduction (AWF):** a per-layer low-rank factorization that directly minimizes input-aware reconstruction error, providing a principled alternative to weight-space SVD.
2. **Optimal rank allocation (ORA):** a global allocator that formulates rank assignment as a budget-constrained optimization problem, solved efficiently via water-filling and yielding provably optimal assignments.

We showed that AFORA outperforms truncated SVD and pruning-based baselines. At a 15% compression ratio, AFORA reduces perplexity on WikiText-2 by 8.36% compared to the best existing method, shortens compression algorithm execution time by up to $9.6\times$, and preserves zero-shot accuracy on reasoning benchmarks at a level comparable to the dense model.

Beyond empirical performance, our analysis emphasizes that activation statistics and optimization-theoretic allocation can transform compression from heuristic design into a systematic optimization problem. AFORA thus provides both a practical tool for efficient LLM deployment and a theoretical foundation for further refinements, such as incorporating richer loss-aware objectives or exploring tighter integration with pruning-based approaches.

In summary, AFORA demonstrates that combining activation-aware factorization with principled global rank allocation offers a viable and theoretically grounded pathway toward efficient, training-free deployment of large language models.

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

## A  OPTIMALITY OF TRUNCATED SVD

The following theorem is a classical result due to Eckart–Young–Mirsky. It formally establishes that truncated SVD gives the optimal rank-$r$ approximation under the Frobenius norm.

**Theorem A.1** (Eckart–Young–Mirsky). *For any matrix $\boldsymbol{W} \in \mathbb{R}^{d_{out} \times d_{in}}$ with singular values $\sigma_1 \geq \sigma_2 \geq \cdots \geq \sigma_{\min(d_{out}, d_{in})}$, the solution to*

$$\min_{\mathrm{rank}(\boldsymbol{X}) \leq r} \quad \|\boldsymbol{W} - \boldsymbol{X}\|_F \tag{23}$$

*is given by*

$$\boldsymbol{X}^\star = \boldsymbol{U}_r \boldsymbol{S}_r \boldsymbol{V}_r^\top, \tag{24}$$

*where $\boldsymbol{U}_r, \boldsymbol{S}_r, \boldsymbol{V}_r$ are the top-$r$ singular components of $\boldsymbol{W}$. Moreover, the approximation error is*

$$\|\boldsymbol{W} - \boldsymbol{X}^\star\|_F^2 = \sum_{i=r+1}^{\min(d_{out}, d_{in})} \sigma_i^2. \tag{25}$$

This establishes truncated SVD as the optimal low-rank solution in the standard Frobenius sense. Our activation-aware formulation (Sec. 3) extends this principle by replacing the weight-space Frobenius norm with a whitened, loss-aware metric.

## B  FROM ACTIVATION-AWARE PROBLEM TO COVARIANCE FORMULATION

We show how the activation-aware objective in equation 2 is equivalent to the covariance formulation in equation 3.

Starting point:

$$\min_{\mathrm{rank}(BA) \leq r} \quad \mathbb{E}_x \Big[ \|Wx - BAx\|_F^2 \Big]. \tag{26}$$

Expand the Frobenius norm:

$$\mathbb{E}_x \Big[ \|Wx - BAx\|_F^2 \Big] = \mathbb{E}_x \Big[ (Wx - BAx)^\top (Wx - BAx) \Big] \tag{27}$$

$$= \mathbb{E}_x \Big[ x^\top (W - BA)^\top (W - BA)x \Big]. \tag{28}$$

Bring expectation inside:

$$= \mathrm{Tr}\Big((W - BA)^\top (W - BA) \, \mathbb{E}[xx^\top]\Big). \tag{29}$$

Define the input covariance $\Sigma_{\mathrm{in}} = \mathbb{E}[xx^\top]$, then

$$\mathbb{E}_x\Big[\|Wx - BAx\|_F^2\Big] = \mathrm{Tr}\Big((W - BA)^\top (W - BA) \, \Sigma_{\mathrm{in}}\Big). \tag{30}$$

Rewriting with the Frobenius norm identity:

$$= \big\|(W - BA)\Sigma_{\mathrm{in}}^{1/2}\big\|_F^2. \tag{31}$$

Therefore the problem is equivalent to

$$\min_{\mathrm{rank}(BA)\leq r} \big\|W\Sigma_{\mathrm{in}}^{1/2} - BA\Sigma_{\mathrm{in}}^{1/2}\big\|_F^2, \tag{32}$$

which is exactly equation 3.

## C  APPENDIX: DERIVATION OF INFORMATION-BALANCED FACTORIZATION

We derive the closed-form scaling for information-balanced factorization. Starting from the truncated SVD factors

$$B_0 = U_r \, \mathrm{diag}(\sqrt{s_{1:r}}), \tag{33}$$

$$A_0 = \mathrm{diag}(\sqrt{s_{1:r}}) \, V_r^\top \Sigma_{\mathrm{in}}^{-1/2}, \tag{34}$$

the column norms of $B_0$ and row norms of $A_0$ may be unbalanced.

Introduce positive scalings $\alpha_i > 0$:

$$B(\alpha) = U_r \, \mathrm{diag}(\alpha_1, \ldots, \alpha_r), \tag{35}$$

$$A(\alpha) = \mathrm{diag}\Big(\frac{s_i}{\alpha_i}\Big) \, V_r^\top \Sigma_{\mathrm{in}}^{-1/2}. \tag{36}$$

For each component $i$, the relevant norms are

$$\|B(\alpha)_{:,i}\|_2 = \alpha_i, \tag{37}$$

$$\|A(\alpha)_{i,:}\|_2 \propto \frac{s_i}{\alpha_i}, \tag{38}$$

$$\|(\Sigma_{\mathrm{in}}^{1/2} V_r)_{:,i}\|_2 = \text{fixed}. \tag{39}$$

Balancing the first two terms against the third gives

$$\alpha_i^\star = \sqrt{s_i \cdot \|(\Sigma_{\mathrm{in}}^{1/2} V_r)_{:,i}\|_2}, \qquad i = 1, \ldots, r. \tag{40}$$

## D  OPTIMALITY OF ORA UNDER PREFIX STRUCTURE

We show that the convex relaxation of the ORA objective (Sec. 4) admits an integral optimum under the prefix structure of singular values. This establishes that the water-filling procedure yields an exact solution when no floors are imposed.

Consider the allocation problem

$$\max_{\{r_m \in \mathbb{Z}_{\geq 0}\}} \sum_{m=1}^{M} \sum_{i=1}^{r_m} v_{m,i} \qquad \text{s.t.} \quad \sum_{m=1}^{M} c_m r_m \leq \mathcal{B}, \tag{41}$$

with per-rank utilities $v_{m,i}$ that are nonincreasing in $i$. The problem is a variant of the multiple-choice knapsack problem (MCKP) (Boyd & Vandenberghe, 2004), where each direction $(m, i)$ is an item with value $v_{m,i}$ and cost $c_m$.

Because singular values are sorted, utilities satisfy

$$v_{m,1} \geq v_{m,2} \geq \cdots, \tag{42}$$

so each block must be taken as a prefix $\{1, \ldots, r_m\}$. Otherwise, replacing a later singular direction with an earlier one increases value at no higher cost.

Relaxing to $r_m \in \mathbb{R}_{\geq 0}$ with multiplier $\lambda \geq 0$ gives the stationarity condition

$$\frac{v_{m,i}}{c_m} \geq \lambda \;\; \text{for } i \leq r_m^\star, \qquad \frac{v_{m,i}}{c_m} < \lambda \;\; \text{for } i > r_m^\star. \tag{43}$$

This is exactly the water-filling rule: keep all directions with density above the global cutoff $\lambda^\star$. Since selections are per-layer prefixes, the solution $r_m^\star$ is automatically integer-valued. If ties occur at the cutoff, multiple integer solutions exist with equal objective value.

When minimum ranks $r_m^{\min}$ are required, the same prefix argument applies after fixing the floors. Feasibility requires

$$\sum_m c_m r_m^{\min} \leq \mathcal{B}. \tag{44}$$

If infeasible, some floors must be reduced. Our heuristic (Sec. 4.5) reduces the floor with the smallest boundary density until feasible. This is practical but not guaranteed globally optimal; more exact strategies (e.g., integer programming) could be applied.

Thus ORA admits an exact integral optimum under the prefix structure, and the water-filling procedure recovers it directly. With floors, optimality holds for the feasible case, while infeasible cases require heuristic handling.

## E    LAYER-WISE WHITENED SPECTRA

We whiten inputs with the estimated second moment and compute singular values per linear module:

$$T_m \;=\; W_m \Sigma_{\text{in},m}^{1/2}, \tag{45}$$
$$T_m \;=\; U_m \operatorname{diag}(s_{m,1}, \ldots, s_{m,d_m}) V_m^\top. \tag{46}$$

We visualize the squared singular values $s_{m,i}^2$ together with the cumulative energy

$$E_m(k) \;=\; \frac{\sum_{i=1}^{k} s_{m,i}^2}{\sum_{i=1}^{d_m} s_{m,i}^2}, \tag{47}$$

which indicates how quickly energy concentrates across ranks. Large early values of $s_{m,i}^2$ (and rapidly rising $E_m$) mean a few directions capture most of the input-aware effect, so low ranks suffice; flatter spectra imply higher ranks are needed. Across layers and modules, the shapes vary substantially, and we use these diagnostics to summarize compression difficulty and guide budgeted allocation (see Fig. 3).

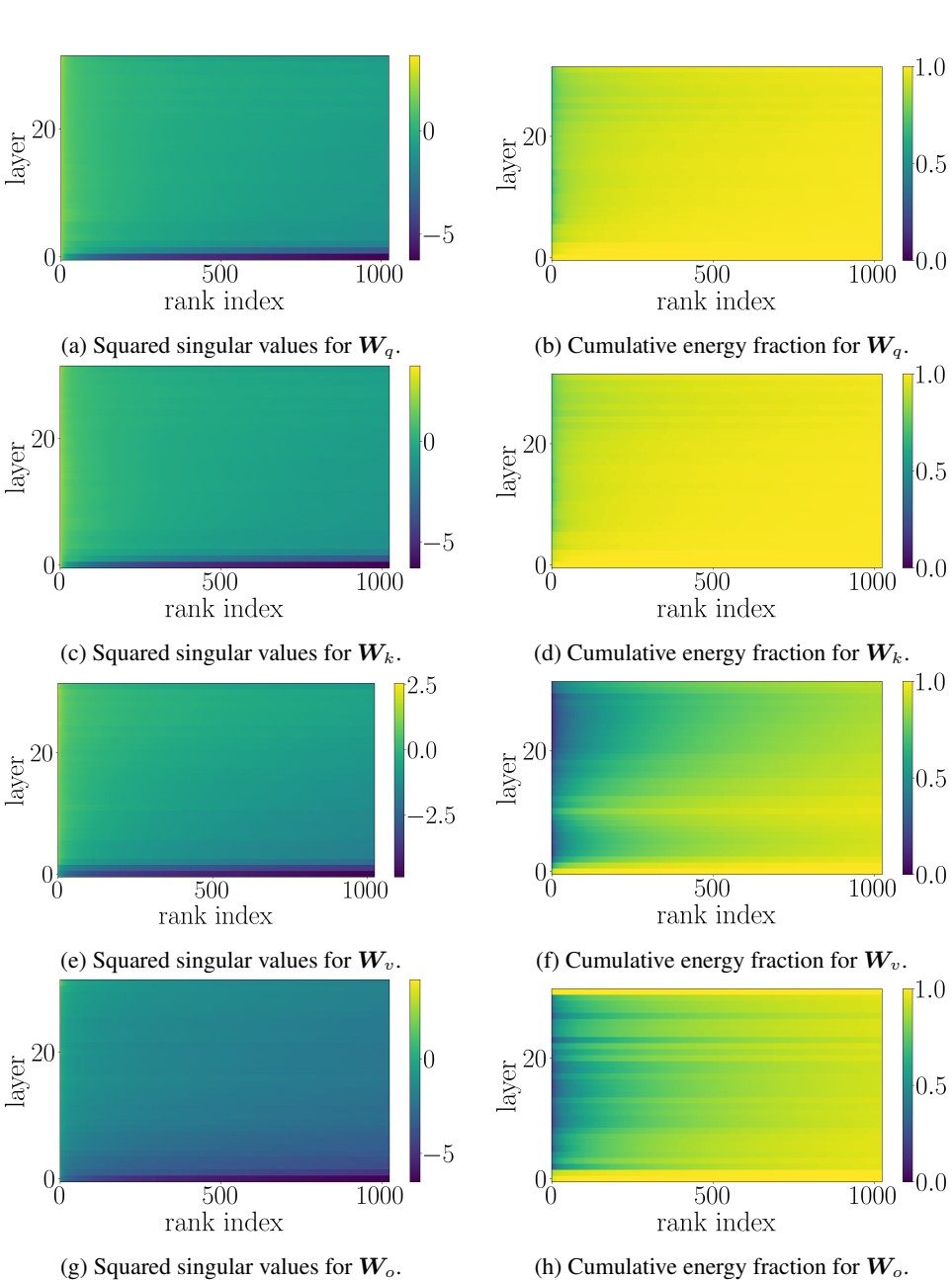

(a) Squared singular values for $\boldsymbol{W}_q$.

(b) Cumulative energy fraction for $\boldsymbol{W}_q$.

(c) Squared singular values for $\boldsymbol{W}_k$.

(d) Cumulative energy fraction for $\boldsymbol{W}_k$.

(e) Squared singular values for $\boldsymbol{W}_v$.

(f) Cumulative energy fraction for $\boldsymbol{W}_v$.

(g) Squared singular values for $\boldsymbol{W}_o$.

(h) Cumulative energy fraction for $\boldsymbol{W}_o$.

Figure 3: **Heatmaps of spectra across modules (attention projections** $q, k, v, o$**).** Each row corresponds to one projection type ($q, k, v, o$ from top to bottom). The left column shows squared singular values $s_{m,i}^2$ across layers (horizontal axis: rank index; vertical axis: layer depth). The right column shows the corresponding cumulative energy curves $E_m(k)$.

# F  ADDITIONAL EXPERIMENTS ON LLaMA-2-7B

## F.1  LANGUAGE MODELING (PPL) VS. BUDGET ON LLaMA-2-7B

Table 5: Language modeling performance at 5% compression ratio.

| Method | Compression Ratio in MHA↑ | PPL(WikiText-2)↓ | PPL(PTB)↓ |
|---|---|---|---|
| Dense | | 10.46 | 128.01 |
| SVD | 0.16 | 94.67 | 400.01 |
| FWSVD | 0.16 | 136.91 | 489.16 |
| ASVD | | **10.35** | **127.45** |
| SVD-LLM | | 13.40 | 151.78 |
| SLEB (prune) | | 11.48 | 130.94 |
| **AWF** | 0.16 | 10.97± 0.05 | 129.48± 0.31 |
| *AFORA* | 0.16 | 10.95± 0.04 | 131.47± 0.58 |

Table 6: Language modeling performance at 10% compression ratio.

| Method | Compression Ratio in MHA↑ | PPL(WikiText-2)↓ | PPL(PTB)↓ |
|---|---|---|---|
| Dense | | 10.46 | 128.01 |
| SVD | 0.31 | 310.16 | 1082.86 |
| FWSVD | 0.31 | 396.18 | 969.46 |
| ASVD | | **11.12** | **127.52** |
| SVD-LLM | | 14.18 | 161.82 |
| SLEB (prune) | | 11.99 | 132.45 |
| **AWF** | 0.31 | 11.58± 0.06 | 139.23± 1.72 |
| *AFORA* | 0.31 | 11.36± 0.05 | 137.39± 0.55 |

## F.2  ZERO-SHOT ACCURACY

Table 7: Zero-shot evaluation at 5% compression ratio on LLaMA-2-7B.

| Method | ARC-E | HellaSwag | PIQA | WinoGrande | CB | OpenbookQA | GSM8K | Avg. |
|---|---|---|---|---|---|---|---|---|
| Dense | 0.71 | 0.70 | 0.79 | 0.68 | 0.48 | 0.28 | 0.13 | 0.54 |
| SVD | 0.46 | 0.45 | 0.60 | 0.58 | 0.46 | 0.21 | 0.00 | 0.39 |
| FWSVD | 0.67 | 0.67 | 0.74 | 0.67 | 0.39 | 0.25 | 0.01 | 0.49 |
| ASVD | 0.67 | 0.69 | 0.76 | **0.70** | 0.34 | **0.28** | 0.09 | 0.50 |
| SVD-LLM | 0.59 | 0.61 | 0.70 | 0.67 | 0.41 | 0.21 | 0.05 | 0.46 |
| SLEB (depth) | 0.65 | **0.70** | 0.72 | 0.65 | 0.30 | 0.26 | 0.02 | 0.47 |
| *AWF* | 0.66 ± 0.01 | 0.63 ± 0.01 | 0.73 ± 0.02 | **0.70** ± 0.04 | 0.59 ± 0.08 | 0.26 ± 0.02 | 0.07 ± 0.02 | 0.52 ± 0.01 |
| *AFORA* | **0.69** ± 0.01 | 0.69 ± 0.01 | **0.79** ± 0.00 | 0.67 ± 0.01 | **0.59** ± 0.02 | 0.26 ± 0.01 | 0.07 ± 0.02 | **0.54** ± 0.00 |

Table 8: Zero-shot evaluation at 10% compression ratio on `LLaMA-2-7B`.

| Method | ARC-E | HellaSwag | PIQA | WinoGrande | CB | OpenbookQA | GSM8K | Avg. |
|---|---|---|---|---|---|---|---|---|
| Dense | 0.71 | 0.70 | 0.79 | 0.68 | 0.48 | 0.28 | 0.13 | 0.54 |
| SVD | 0.28 | 0.46 | 0.52 | 0.52 | 0.36 | 0.12 | 0.01 | 0.32 |
| FWSVD | 0.51 | 0.60 | 0.69 | 0.65 | 0.44 | 0.25 | 0.00 | 0.45 |
| ASVD | 0.66 | **0.68** | 0.75 | **0.74** | 0.48 | **0.31** | **0.09** | **0.53** |
| SVD-LLM | 0.56 | 0.62 | 0.70 | 0.70 | 0.41 | 0.21 | 0.03 | 0.46 |
| SLEB (depth) | 0.59 | 0.60 | 0.69 | 0.59 | 0.45 | 0.23 | 0.01 | 0.45 |
| *AWF* | 0.63 ± 0.02 | 0.61 ± 0.02 | **0.78** ± 0.01 | 0.68 ± 0.02 | **0.56** ± 0.02 | 0.24 ± 0.02 | 0.03 ± 0.02 | 0.52 ± 0.01 |
| *AFORA* | **0.71** ± 0.03 | 0.65 ± 0.02 | **0.78** ± 0.02 | 0.68 ± 0.01 | **0.56** ± 0.05 | 0.24 ± 0.01 | 0.03 ± 0.00 | 0.52 ± 0.01 |

# G   ADDITIONAL EXPERIMENTS ON `LLaMA-3-8B`

## G.1   LANGUAGE MODELING (PPL) VS. BUDGET ON `LLaMA-3-8B`

Table 9: Language modeling performance at 2.5% compression ratio.

| Method | Compression Ratio in MHA↑ | PPL(WikiText-2)↓ | PPL(PTB)↓ |
|---|---|---|---|
| Dense | | 14.16 | 90.90 |
| SVD | 0.35 | 24.63 | 133.51 |
| FWSVD | 0.35 | 50425.62 | 21320.24 |
| ASVD | | **14.33** | **93.31** |
| *AFORA* | 0.15 | 14.59± 0.02 | 95.91± 0.44 |

Table 10: Language modeling performance at 5% compression ratio.

| Method | Compression Ratio in MHA↑ | PPL(WikiText-2)↓ | PPL(PTB)↓ |
|---|---|---|---|
| Dense | | 14.16 | 90.90 |
| SVD | 0.35 | 24.63 | 133.51 |
| FWSVD | 0.35 | 50425.62 | 21320.24 |
| ASVD | | 15.43 | **98.01** |
| *AFORA* | 0.30 | **14.89**± 0.04 | 98.15± 0.40 |

Table 11: Language modeling performance at 7.5% compression ratio.

| Method | Compression Ratio in MHA↑ | PPL(WikiText-2)↓ | PPL(PTB)↓ |
|---|---|---|---|
| Dense | | 14.16 | 90.90 |
| SVD | 0.45 | 85.72 | 436.64 |
| FWSVD | 0.45 | 32150.11 | 24532.78 |
| ASVD | | 18.17 | 107.56 |
| *AFORA* | 0.45 | **15.30**± 0.04 | **105.45**± 0.64 |

## G.2 ZERO-SHOT ACCURACY

Table 12: Zero-shot evaluation at 2.5% compression ratio on `LLaMA-3-8B`.

| Method | ARC-E | HellaSwag | PIQA | WinoGrande | CB | OpenbookQA | GSM8K | Avg. |
|---|---|---|---|---|---|---|---|---|
| Dense | 0.82 | 0.68 | 0.82 | 0.74 | 0.63 | 0.31 | 0.53 | 0.65 |
| SVD | 0.69 | 0.65 | 0.76 | 0.70 | 0.41 | 0.20 | 0.07 | 0.50 |
| FWSVD | 0.33 | 0.32 | 0.55 | 0.50 | 0.41 | 0.08 | 0.02 | 0.32 |
| ASVD | **0.80** | 0.68 | 0.82 | **0.76** | **0.54** | **0.45** | **0.50** | **0.65** |
| *AFORA* | 0.76 ± 0.01 | **0.70** ± 0.01 | **0.83** ± 0.01 | **0.76** ± 0.01 | 0.52 ± 0.02 | 0.30 ± 0.01 | 0.37 ± 0.04 | 0.61 ± 0.01 |

Table 13: Zero-shot evaluation at 5% compression ratio on `LLaMA-3-8B`.

| Method | ARC-E | HellaSwag | PIQA | WinoGrande | CB | OpenbookQA | GSM8K | Avg. |
|---|---|---|---|---|---|---|---|---|
| Dense | 0.82 | 0.68 | 0.82 | 0.74 | 0.63 | 0.31 | 0.53 | 0.65 |
| SVD | 0.69 | 0.65 | 0.76 | 0.70 | 0.41 | 0.20 | 0.07 | 0.50 |
| FWSVD | 0.33 | 0.32 | 0.55 | 0.50 | 0.41 | 0.08 | 0.02 | 0.32 |
| ASVD | **0.77** | **0.70** | **0.83** | **0.79** | 0.48 | **0.36** | **0.50** | **0.63** |
| *AFORA* | 0.74 ± 0.02 | 0.66 ± 0.02 | 0.82 ± 0.02 | 0.76 ± 0.01 | **0.49** ± 0.02 | 0.30 ± 0.01 | 0.25 ± 0.04 | 0.58 ± 0.00 |

Table 14: Zero-shot evaluation at 7.5% compression ratio on `LLaMA-3-8B`.

| Method | ARC-E | HellaSwag | PIQA | WinoGrande | CB | OpenbookQA | GSM8K | Avg. |
|---|---|---|---|---|---|---|---|---|
| Dense | 0.82 | 0.68 | 0.82 | 0.74 | 0.63 | 0.31 | 0.53 | 0.65 |
| SVD | 0.60 | 0.53 | 0.75 | 0.62 | 0.38 | 0.20 | 0.03 | 0.44 |
| FWSVD | 0.25 | 0.33 | 0.49 | 0.52 | 0.41 | 0.07 | 0.00 | 0.30 |
| ASVD | **0.75** | **0.68** | **0.79** | **0.77** | **0.54** | 0.30 | **0.26** | **0.58** |
| *AFORA* | 0.73 ± 0.02 | 0.66 ± 0.02 | 0.78 ± 0.01 | 0.74 ± 0.02 | 0.41 ± 0.00 | **0.31** ± 0.02 | 0.14 ± 0.03 | 0.54 ± 0.01 |

# H ADDITIONAL EXPERIMENTS ON QWEN-3-8B

## H.1 LANGUAGE MODELING (PPL) VS. BUDGET ON QWEN-3-8B

Table 15: Language modeling performance at 2.5% compression ratio.

| Method | Compression Ratio in MHA↑ | PPL(WikiText-2)↓ | PPL(PTB)↓ |
|---|---|---|---|
| Dense | | 18.47 | 116.95 |
| SVD | 0.35 | 33.76 | 273.60 |
| FWSVD | 0.35 | 166445.64 | 116773.84 |
| ASVD | | **18.58** | **109.15** |
| *AFORA* | 0.14 | 19.62± 0.03 | 119.69± 1.19 |

Table 16: Language modeling performance at 5% compression ratio.

| Method | Compression Ratio in MHA↑ | PPL(WikiText-2)↓ | PPL(PTB)↓ |
|---|---|---|---|
| Dense | | 18.47 | 116.95 |
| SVD | 0.35 | 33.76 | 273.60 |
| FWSVD | 0.35 | 166445.64 | 116773.84 |
| ASVD | | **20.00** | **118.02** |
| *AFORA* | 0.28 | 20.40± 0.07 | 130.11± 4.81 |

Table 17: Language modeling performance at 7.5% compression ratio.

| Method | Compression Ratio in MHA↑ | PPL(WikiText-2)↓ | PPL(PTB)↓ |
|---|---|---|---|
| Dense | | 18.47 | 116.95 |
| SVD | 0.42 | 114.51 | 6053.02 |
| FWSVD | 0.42 | 4740616.01 | 3600659.72 |
| ASVD | | 21.77 | **126.90** |
| *AFORA* | 0.42 | **20.78**± 0.10 | 137.17± 3.50 |

## H.2 ZERO-SHOT ACCURACY

Table 18: Zero-shot evaluation at 2.5% compression ratio on `QWEN-3-8B`.

| Method | ARC-E | HellaSwag | PIQA | WinoGrande | CB | OpenbookQA | GSM8K | Avg. |
|---|---|---|---|---|---|---|---|---|
| Dense | 0.82 | 0.67 | 0.82 | 0.78 | 0.73 | 0.30 | 0.91 | 0.72 |
| SVD | 0.74 | 0.62 | 0.77 | 0.64 | 0.57 | 0.30 | 0.52 | 0.59 |
| FWSVD | 0.37 | 0.43 | 0.59 | 0.42 | 0.27 | 0.19 | 0.02 | 0.33 |
| ASVD | 0.79 | **0.64** | **0.84** | **0.77** | 0.16 | **0.32** | 0.62 | 0.59 |
| *AFORA* | **0.80** ± 0.02 | 0.63 ± 0.01 | 0.79 ± 0.02 | 0.75 ± 0.02 | **0.76** ± 0.04 | 0.29 ± 0.01 | **0.81** ± 0.01 | **0.69** ± 0.01 |

Table 19: Zero-shot evaluation at 5% compression ratio on `QWEN-3-8B`.

| Method | ARC-E | HellaSwag | PIQA | WinoGrande | CB | OpenbookQA | GSM8K | Avg. |
|---|---|---|---|---|---|---|---|---|
| Dense | 0.82 | 0.67 | 0.82 | 0.78 | 0.73 | 0.30 | 0.91 | 0.72 |
| SVD | 0.74 | 0.62 | 0.77 | 0.64 | 0.57 | 0.30 | 0.52 | 0.59 |
| FWSVD | 0.37 | 0.43 | 0.59 | 0.42 | 0.27 | 0.19 | 0.02 | 0.33 |
| ASVD | **0.79** | 0.62 | **0.83** | **0.73** | 0.39 | **0.31** | 0.58 | 0.61 |
| *AFORA* | 0.78 ± 0.02 | **0.63** ± 0.00 | 0.77 ± 0.03 | 0.71 ± 0.02 | **0.75** ± 0.03 | 0.27 ± 0.01 | **0.71** ± 0.05 | **0.66** ± 0.02 |

Table 20: Zero-shot evaluation at 7.5% compression ratio on `QWEN-3-8B`.

| Method | ARC-E | HellaSwag | PIQA | WinoGrande | CB | OpenbookQA | GSM8K | Avg. |
|---|---|---|---|---|---|---|---|---|
| Dense | 0.82 | 0.67 | 0.82 | 0.78 | 0.73 | 0.30 | 0.91 | 0.72 |
| SVD | 0.48 | 0.61 | 0.69 | 0.60 | 0.57 | 0.23 | 0.00 | 0.45 |
| FWSVD | 0.40 | 0.36 | 0.58 | 0.44 | 0.48 | 0.12 | 0.00 | 0.34 |
| ASVD | **0.81** | **0.62** | **0.83** | **0.75** | 0.48 | **0.32** | 0.31 | 0.59 |
| *AFORA* | 0.78 ± 0.02 | 0.60 ± 0.01 | 0.77 ± 0.03 | 0.68 ± 0.02 | **0.73** ± 0.05 | 0.27 ± 0.01 | **0.59** ± 0.13 | **0.63** ± 0.01 |

# I    ADDITIONAL ABLATION RESULTS: ASVD COMBINED WITH ORA

To further isolate the contribution of ORA, we also evaluate the combination **ASVD + ORA**. This ablation helps clarify whether ORA provides value even when paired with a different activation-aware factorization method. Table 21 and Table 22 report results on `LLaMA-3-8B` at a compression ratio of 0.15. ORA yields small improvements over ASVD in zero-shot accuracy, although the perplexity can be higher, especially when utilities are derived from whitened operators (ORA$^{\ddagger}$). These observations are consistent with the design of ASVD, which selects ranks by directly evaluating every possible choice, whereas ORA chooses ranks by solving a simple optimization problem. As a result, ASVD's allocation tends to achieve lower perplexity, while ORA is far more computationally efficient and can still provide modest gains in generalization metrics.

Table 21: Perplexity of ASVD with and without ORA on `LLaMA-3-8B` (compression ratio 0.05).

| Method | PPL(Wikitext-2) | PPL(PTB) |
|---|---|---|
| ASVD | 14.33 | 98.01 |
| ASVD + ORA | 21.36± 0.22 | 152.45 ± 1.39 |

Table 22: Zero-shot accuracy of ASVD with and without ORA on `LLaMA-3-8B` (compression ratio 0.05).

| Method | ARC-E | HellaSwag | PIQA | WinoGrande | CB | OpenbookQA | GSM8K | Avg. |
|---|---|---|---|---|---|---|---|---|
| ASVD | 0.77 | 0.70 | 0.83 | 0.79 | 0.48 | 0.36 | 0.50 | 0.63 |
| **ASVD + ORA** | 0.82 ± 0.00 | 0.68 ± 0.00 | 0.82 ± 0.00 | 0.74 ± 0.00 | 0.63 ± 0.00 | 0.31 ± 0.00 | 0.53 ± 0.00 | 0.65 ± 0.00 |

Overall, these results indicate that ORA remains compatible with alternative activation-aware factorizations such as ASVD. While ASVD's exhaustive per-layer search often yields lower perplexity, ORA provides a significantly cheaper mechanism for rank allocation and can still improve zero-shot performance relative to ASVD alone.

# J    RANK-ALLOCATION PATTERNS: ORA VS. ASVD

Figure 4 visualizes the rank distributions produced by ORA and ASVD. Recall that AFORA applies compression only to the attention projections, whereas ASVD also includes the MLP blocks. Within the attention layers, both methods show broadly similar prioritization patterns, with small differences due to whitening.

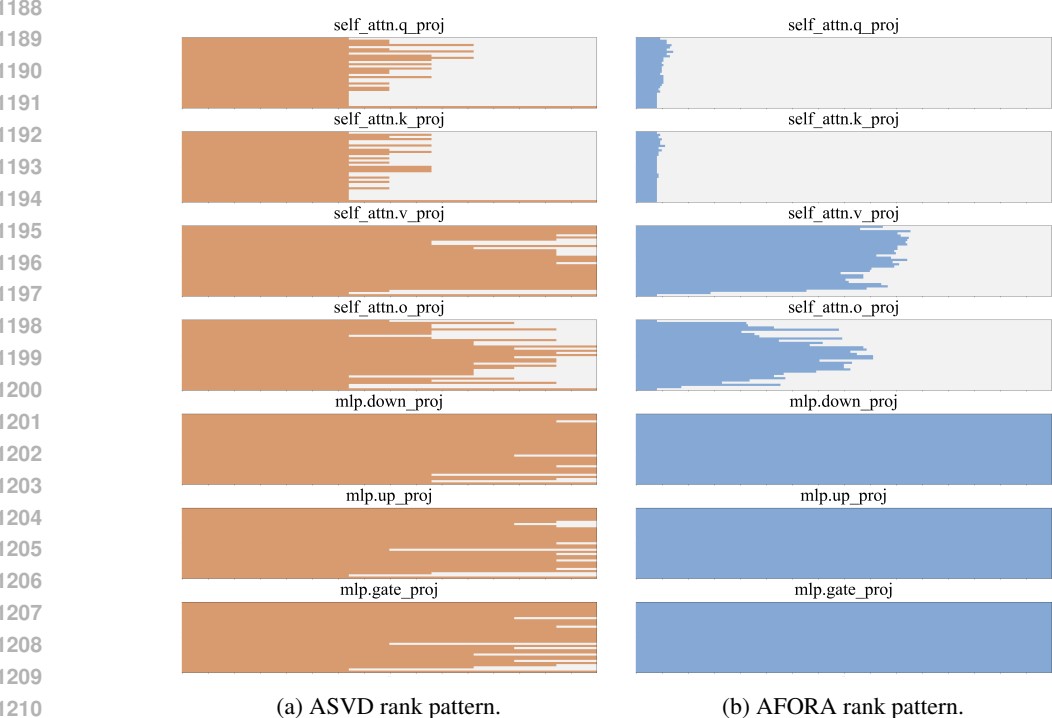

(a) ASVD rank pattern.         (b) AFORA rank pattern.

Figure 4: **Comparison of rank allocation: ASVD vs. AFORA.** ASVD uses exhaustive per-layer search, while AFORA derives rank assignments from activation-aware utilities and a global budget.

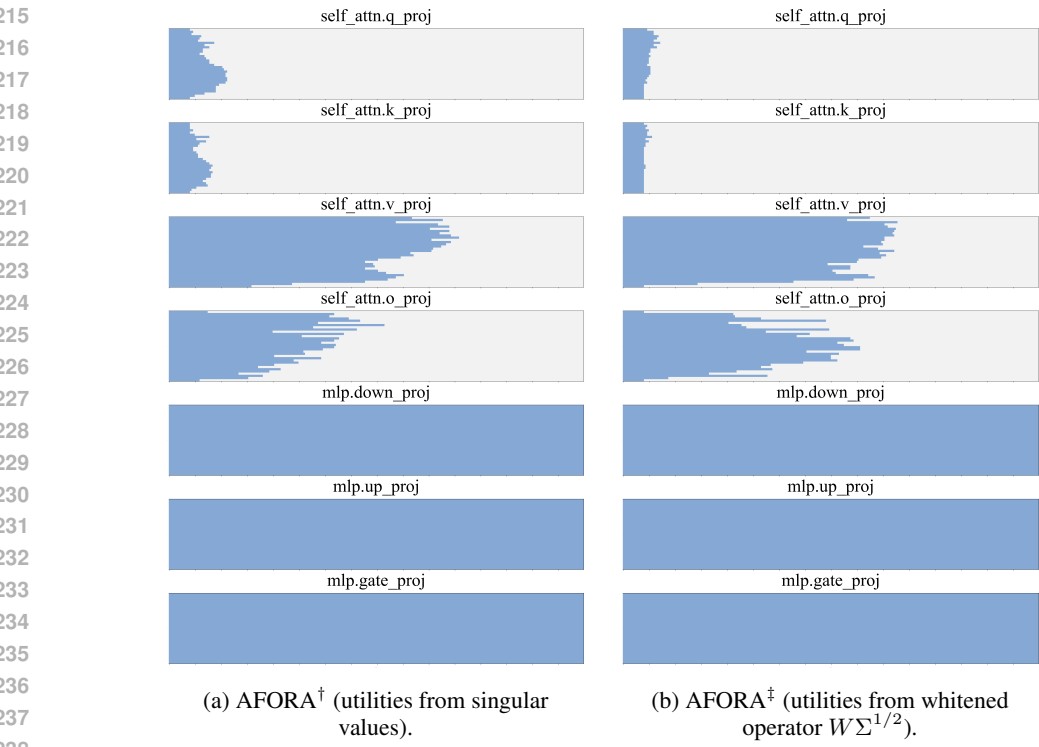

(a) AFORA[†] (utilities from singular values).      (b) AFORA[‡] (utilities from whitened operator $W\Sigma^{1/2}$).

Figure 5: **Comparison of AFORA variants.** AFORA[†] computes utilities from the singular values of $W$, while AFORA[‡] uses the whitened operator $W\Sigma^{1/2}$, leading to slightly different rank distributions under the same compression budget.

