# OpenReview forum: "AFORA: Activation-aware Factorization with Optimal Rank Allocation for Training-free LLM Compression"
_ICLR.cc/2026/Conference — Submitted to ICLR 2026_

### Official Review · Reviewer_vUu2 · 2025-10-29

**Soundness:** 3
**Presentation:** 3
**Contribution:** 3
**Rating:** 6
**Confidence:** 3

**Summary:**

This paper introduces AFORA, a training-free compression framework for LLMs. The method combines two key ideas: (1) Activation-aware Weight Factorization (AWF), which performs low-rank decomposition by accounting for input activation, and (2) Optimal Rank Allocation (ORA), a global optimization strategy that allocates ranks across layers. The paper provides strong theoretical grounding, deriving the method from loss approximations under the K-FAC assumption, and proving optimality of the global allocation. Experiments on LLaMA-2-7B demonstrate that AFORA outperforms existing training-free compression baselines.

**Strengths:**

- The paper is technically rigorous, with clear mathematical derivations and proofs connecting activation-aware factorization to second-order loss approximations.

- The paper is well-written, conceptually coherent.

- Empirical performance is strong, albeit demonstrated on a single model.

**Weaknesses:**

- The evaluation is limited to a single model (LLaMA-2-7B), leaving open the question of generalization to other model families or scales. As a compression technique is meant to be model-agnostic, demonstrating consistency across architectures would significantly strengthen the empirical claims.

- Although the method is described as hardware-friendly, there is no profiling on diverse hardware or latency benchmarks.

- The method’s behavior under extreme compression ratios (<5%) or its compatibility with quantization/pruning hybrids is not explored.

**Questions:**

- How sensitive is AFORA to the quality or size of the calibration dataset used to estimate Σ_in? Would smaller or domain-shifted calibration streams degrade performance?

- Can the authors provide results on other model families to demonstrate architectural generality?

- What is the runtime or memory overhead of computing Σ_in and performing SVDs at large scales ?

- How does AFORA interact with quantization or pruning if applied sequentially ?

---

> ### Author Response · Authors · 2025-11-21
> **Official Comment by Authors**
>
> # Response to Reviewer vUu2
>
> Thank you for the constructive feedback and for raising several points that were not fully addressed in the original submission.
>
> ---
>
> ## **Q1**. Sensitivity to calibration data (Σ_in)
>
> > Q1. How sensitive is AFORA to the quality or size of the calibration dataset used to estimate Σ_in? Would smaller or domain-shifted calibration streams degrade performance?
>
> We ran an additional analysis to understand how sensitive AFORA is to the size of the calibration set used to estimate Σ_in. The [attached figure](https://ibb.co/LD1jDc82) (linked externally due to space constraints) plots the perplexity of the compressed model as a function of the calibration size. We observed that around 100 batches (batch size = 1) were already sufficient for stable performance. These results have been included in the revised manuscript.
>
> ---
>
> ## **W1Q2**. Generality across model families
>
> > Q2. Can the authors provide results on other model families to demonstrate architectural generality?
>
> As suggested, we extended the experiments to other architectures. We are running AFORA on LLaMA-3 and Qwen, and the preliminary results from LLaMA-3 follow the same trend as those of LLaMA-2-7B. These additional experiments have been included to support the architectural generality of the method.
>
> ### Table 1. Perplexity on LLaMA-3-8B
>
> Note: LLaMA-3 projection layers' different shapes impose a shared maximum rank(1024) in AWF, SVD.
> Note: ASVD allocates rank to the MLP layers.
> | | #Params | Compression Ratio | Compression Ratio in MHA | PPL(Wikitext-2) | PPL(PTB) |
> |-|-|-|-|-|-|
> |Dense|8030261248| 0| |14.16| 90.90 |
> |SVD|7560499200| 0.06 | 0.35 | 24.63 | 133.51 |
> |ASVD| 7628748186|0.05| |15.43|98.01|
> |AFORA|7628748186|0.05|0.30|14.89 ± 0.04|98.15 ± 0.72|
> ### Table 2. Zero-shot results on LLaMA-3-8B
> | |ARC-E|HellaSwag|PIQA|WinoGrande|CB|OpenbookQA|GSM8K|AVG
> |-|-|-|-|-|-|-|-|-|
> |Dense|0.82|0.68|0.82|0.74|0.63|0.31|0.53|0.65|
> |SVD|0.69|0.65|0.76|0.70|0.41|0.20|0.07|0.50|
> |ASVD|0.77|0.70|0.83|0.79|0.48|0.36|0.50|0.63|
> |AFORA|0.74 ± 0.02|0.66 ± 0.02|0.82 ± 0.02|0.76 ± 0.01|0.49 ± 0.02|0.30 ± 0.01|0.25 ± 0.04|0.58 ± 0.00|
>
> ---
>
> ## **Q3**. Overhead of estimating Σ_in and performing SVD at scale
>
> > Q3. What is the runtime or memory overhead of computing Σ_in and performing SVDs at large scales ?
>
> The computational cost of estimating Σ_in is proportional to a forward pass over the calibration set, since the covariance is computed from activations collected during inference. This part scales linearly with data size. The SVD itself is performed once per linear projection layer, similar to other SVD-based compression methods. While we do not have exact profiling numbers across different scales yet, we have added a short discussion in the revised manuscript and provide an approximate breakdown of the overhead relative to standard SVD-based approaches.
>
> ---
>
> ## **Q4**. Interaction with pruning and quantization
>
> > Q4. How does AFORA interact with quantization or pruning if applied sequentially?
>
> We also explored how AFORA behaves when combined with pruning or quantization:
>
> **Pruning after AFORA.**
> If AFORA assigns very small ranks to all projection layers inside a particular block, the block essentially carries little information. In that case, pruning the entire block may be reasonable, and AFORA can help identify which blocks are candidates for removal.
>
> **AFORA after pruning.**
> If block-level pruning is applied first, AFORA can redistribute the freed-up rank budget more effectively across the remaining layers. This can potentially improve the overall parameter efficiency.
>
> **Quantization after AFORA.**
> AWF includes an “information-balanced” factorization, which reduces large column-norm outliers in the low-rank factors. Lowering outliers generally helps quantization by reducing error amplification, so AFORA can make downstream quantization more stable.
>
> **AFORA after quantization.**
> This ordering is less favorable. Quantization noise distorts the weight matrices before AFORA computes its utilities and sensitivity metrics, which can propagate into the ORA allocation and lead to sub-optimal rank decisions. For this reason, AFORA is better applied before quantization.
>
> ---
>
> Please let us know if any additional clarification would be helpful. We appreciate the reviewer’s comments and have incorporated these analyses into the revised manuscript.

---

> ### Comment · Reviewer_vUu2 · 2025-11-23
> **Acknowledgement of Rebuttal**
>
> Thank you for the authors' additional experiments during the rebuttal period. The conducted experiments, to a large extent, satisfactorily address my questions.
>
> However, I still note only experiments on Llama family. To show generalization, other model 'families' must also be incorporated. I am also unable to see the revised manuscript.

---

> > ### Author Response · Authors · 2025-11-26
> > **Update on Additional Experiments and Revised Manuscript**
> >
> > Thank you for the follow-up comment.
> >
> > We apologize for the slight delay — the additional experiments on other model families (Qwen) required more time to run than expected. The revised manuscript has now been uploaded.
> >
> > Please let us know if anything requires further clarification.

---

> > > ### Comment · Reviewer_vUu2 · 2025-11-27
> > >
> > > Thank you for the experiments, I am satisfied with the rebuttal and shall keep my positive rating.

---

### Official Review · Reviewer_WoAZ · 2025-10-30

**Soundness:** 3
**Presentation:** 3
**Contribution:** 2
**Rating:** 6
**Confidence:** 4

**Summary:**

This paper introduces AFORA, a training-free LLM compression framework that uses low-rank factorization to reduce model size. AFORA has two main components:
- Activation-aware Weight Factorization (AWF): A closed-form method that approximates weight matrices by minimizing output error, considering the input activation distribution. Unlike standard SVD, which focuses on weight reconstruction, AWF preserves task-relevant information.
- Optimal Rank Allocation (ORA): A global strategy that assigns different ranks to each layer to minimize overall error under a fixed parameter budget. It uses a water-filling algorithm to find the globally optimal rank distribution.

On LLaMA-2-7B, AFORA outperforms baselines like SVD and ASVD at various compression ratios. At 15% compression, it significantly lowers perplexity, reduces compression time by up to 9.6×, and achieves a 1.21× inference speedup while maintaining competitive zero-shot accuracy.

**Strengths:**

- AFORA's key innovation is its principled combination of activation-aware factorization (AWF) and globally optimal rank allocation (ORA). This moves beyond heuristic approaches by providing a closed-form, loss-aware solution for factorization and a theoretically grounded method for budget allocation.
- ORA exhibits greater elegance than the rank searching method in ASVD, owing to its robust theoretical background.
- AFORA addresses the need for efficient LLM deployment. Its training-free nature avoids costly fine-tuning, and its hardware-friendly output (dense matrices) ensures broad compatibility. It demonstrates an good balance between performance and efficiency.

**Weaknesses:**

- Experiments are confined to LLaMA-2-7B. The paper lacks evidence of AFORA's effectiveness on other model architectures (e.g., Qwen), scales, or fine-tuned variants (e.g.,Chat version), which limits the claims of its general applicability.
- AFORA achieves the same average zero-shot accuracy but much lower ppl as ASVD at 15% compression. The paper does not analyze why these differences occur, making it difficult to understand the unique benefits of AFORA's design.

**Questions:**

- AWF vs. SVD-LLM: Could you provide a quantitative comparison between AWF and SVD-LLM on LLaMA-2-7B, including perplexity, compression time, and numerical stability? This would clarify AWF's advantages.
- Zero-Shot Parity with ASVD: Can you provide a deeper analysis of the zero-shot results? For instance, why does AFORA excel on certain tasks (e.g., CB) but lag on others (e.g., GSM8K) compared to ASVD? Does this relate to layer-specific error patterns?
- ORA vs. ASVD Rank Allocation: Could you visualize the rank distributions produced by ORA and ASVD across layers? Do both methods prioritize the same layers (e.g., attention vs. MLP)? Explaining any differences would help validate ORA's claimed optimality.

---

> ### Author Response · Authors · 2025-11-21
> **Official Comment by Authors**
>
> Thank you for the thoughtful review and for highlighting several aspects that needed more analysis. Below we address each point in order.
>
> ---
>
> ## **W1**. Experiments on more recent and diverse models
>
> > W1. Experiments are confined to LLaMA-2-7B ...
>
> We agree that evaluating only on LLaMA-2-7B was a limitation of the initial submission. We have extended the experiments to LLaMA-3 and Qwen. The overall trend remains consistent across architectures, supporting the general applicability of AFORA.
>
> ---
>
> ## **W2**. Understanding the gap between ASVD and AFORA
>
> > W2. AFORA achieves the same average zero-shot accuracy but much lower ppl as ASVD ...
>
> Your comment about the performance differences was helpful. We realized that the original draft did not sufficiently explain why AFORA and ASVD behave differently.
>
> ASVD assigns ranks through an exhaustive grid search. Each layer is tested at 0.1-interval retention ratios, and the perplexity drop is measured. However, even this exhaustive procedure cannot guarantee optimality:
> (1) the grid is relatively coarse, and
> (2) the search still cannot consider all joint combinations across layers.
>
> AFORA, on the other hand, uses a utility function that reflects layer importance more directly, and the rank allocation is computed in a single optimization step. Because of this, it is possible that AFORA does not strictly outperform ASVD in every case. But the fact that AFORA—without an exhaustive search—achieves results comparable to ASVD suggests that the utility formulation captures the layer sensitivities more effectively and provides a more efficient alternative.
>
> ---
>
> ## **Q1**. Quantitative comparison: AWF vs. SVD-LLM
>
> > Q1. AWF vs. SVD-LLM: Could you provide a quantitative comparison between AWF and SVD-LLM on LLaMA-2-7B, including perplexity, compression time, and numerical stability? This would clarify AWF's advantages.
>
> We added a comparison between AWF and SVD-LLM.
> The table below summarizes the results on LLaMA-2-7B:
>
> ### Table A. Comparison on LLaMA-2-7B at Compression Ratio 0.15
> |Method|Wikitext-2 PPL|PTB PPL|Zero-shot Acc.|Compression Time|
> |-|-|-|-|-|
> |SVD-LLM|15.12|174.37|0.46|963 sec|
> |AWF|12.88|162.26|0.48|1127 sec|
>
> In our experiments, AWF consistently produced lower perplexity and higher zero-shot accuracy. The compression time (measured with wall-clock time) was similar between the two.
>
> Conceptually, the two methods are related and complementary. For instance, SVD-LLM shows that the squared loss is proportional to the sum of squared singular values, which provides a natural justification for using squared singular values as utilities in ORA. SVD-LLM also performs SVD on \( W S \), where \( S \) comes from a Cholesky-based whitening. This plays a similar role to applying \( \Sigma^{1/2} \) in AWF. We believe the revised manuscript benefites from explicitly discussing these connections.
>
> ---
>
> ## **Q2**. Analysis of zero-shot differences between ASVD and AFORA
>
> > Q2. Zero-Shot Parity with ASVD: Can you provide a deeper analysis of the zero-shot results? ...
>
> We added a more detailed comparison:
>
> ### Table A. Comparison on LLaMA-2-7B at Compression Ratio 0.15
> |Method|ARC-E|HellaSwag|PIQA|Winogrande|CB|OpenBookQA|GSM8K|
> |-|-|-|-|-|-|-|-|
> |ASVD|0.62|0.65|0.71|0.69|0.45|0.28| 0.03|
> |AFORA|0.61|0.64|0.74|0.66|0.52|0.23|0.01|
>
> From these results, we observe a small but consistent pattern: AFORA tends to perform better on truth-judgment tasks such as CB, whereas ASVD sometimes holds an edge on more specialized or multi-step reasoning benchmarks such as GSM8K. This observation is now reflected in the revised manuscript.
>
> ---
>
> ## **Q3**. Rank-allocation patterns: ORA vs. ASVD
>
> > Q3. ORA vs. ASVD Rank Allocation: Could you visualize the rank distributions produced by ORA and ASVD across layers? ...
>
> The reviewer’s question about how ORA and ASVD distribute ranks was very helpful. We prepared [visualizations](https://ibb.co/84zxTy0j) comparing the rank assignments (linked externally for image attachments) and summarize the main observations here.
>
> First, AFORA compresses only the four attention projections, while ASVD also includes the MLP projections, so the sets of candidate layers differ.
>
> From the visualizations, we observe:
>
> - ASVD generally reduces the MLP layers less than the attention layers, suggesting that these MLP projections are more sensitive.
> - Within attention, the overall prioritization patterns are similar between ASVD and ORA:
>   - ASVD: MLP > V > O >> Q ≈ K
>   - ORA: V > O >> Q ≈ K
> - When applying AWF (using \( W \Sigma^{1/2} \)) instead of directly applying SVD on W, the general tendencies remain similar, although there are small shifts in the exact ranks. This reflects the subtle difference in how whitening affects layer sensitivities.
>
> These results have been added in the appendix section.
>
> ---
>
> We appreciate the reviewer’s constructive questions. Please let us know if any other clarification would be helpful.

---

### Official Review · Reviewer_hHNA · 2025-10-31

**Soundness:** 2
**Presentation:** 2
**Contribution:** 1
**Rating:** 2
**Confidence:** 3

**Summary:**

The paper introduces a training-free low-rank compression method (AFORA) for LLMs, built upon two ideas.
1. Activation-aware weight factorization (AWF): Instead of applying SVD directly to each weight matrix, the method uses the empirical covariance of real activations collected from a calibration dataset during the weight factorization and low-rank approximation.
2. Optimal rank allocation (ORA): Under a fixed total storage budget, the method assigns different numbers of ranks to different layers via a water-filling procedure.

**Strengths:**

In terms of originality, this paper proposes a new method (AFORA) that compresses the LLMs, which incorporates the covariance information of the input activations into a closed-form low-rank approximation of the weight matrices.

In terms of quality, the proposed method is mathematically grounded. Both components (AWF and ORA) follow directly from minimizing a clear objective rather than relying on heuristics.

In terms of clarity, the motivations, mathematical derivations, and method descriptions are clear and easy to follow.

In terms of significance, the evaluations show better or similar results than other low-rank approximation methods in both perplexity and zero-shot benchmark accuracies. The method is also efficient and fast to run.

**Weaknesses:**

My main concern is the evaluations, which are very confusing to me.

1. The definition of compression ratio seems inconsistent across the paper.

- The compression ratio is defined in Line 357 as the fraction of parameters retained relative to the dense model.
The ratio columns in Tables 1, 5, and 6 show 85%, 95%, and 90%, with the dense model being 100%, which is consistent with this definition.
However, the captions of Tables 1, 5, and 6, the ratio columns of Tables 3 and 4, the footnotes of Tables 1 and 2, as well as everywhere mentioning the ratio numbers in the main text (including on Line 357 right in front of the definition) use 5%, 10%, and 15%, which is in fact one minus the compression ratio.

- Only the attention projection layers (query, key, value, and output, as described on Line 323) are compressed.
The feed-forward projection layers (gate, up, and down), which take about $\frac{4096 \times 11008 \times 3}{4096 \times 11008 \times 3 + 4096 \times 4096 \times 4} \approx 66.8\\%$ of weights in LLaMA-2-7B, are preserved.
The embeddings, prediction head, and layer norms are also preserved.
The authors should specify whether the compression ratio is computed with respect to the whole model or only the attention layers.
If it is computed only with respect to the attention layers, removing 15% (the maximum ratio tested in the paper) of the attention weights will result in only 5% weight deduction for the whole model, which is not significant.

- I do not understand the last sentence on Line 323: "the same compression ratio is applied across the full model using our global allocator".
It is clear that different layers have different compression ratios using the global rank allocator.

2. Table 4 is not clear.
On Line 101, the reduction in the parameter count and the FLOPs are proportional, while the number of FLOPs reduces much faster than the parameters in Table 4.
The author should fix the definition of compression ratio and better define the FLOP counts, like whether the FLOPs include the attention calculations that increase quadratically with respect to the sequence length.
The calculation method of the speedup is not specified in the paper.
The authors should also mention the input size (batch size and sequence length), as the speedup is generally very sensitive to it.

3. The method seems to be restricted to square weight matrices in practice, while the theoretical part does not impose this restriction.
The evaluations are only done on a small and old LLaMA-2-7B model that has four square weight matrices (the attention projections) in each block.
Note that in many recent LLMs, e.g., Llama-3 series, only query and output layers have square weights (key and value layers become non-square due to head sharing).

4. The baseline evaluations are limited.
The results are primarily compared against other SVD-based methods rather than the more widely used LLM compression techniques, such as the quantization and pruning methods described in Section 2.3, making it difficult to assess the method's relative advantages in practical deployment scenarios.

**Questions:**

1. Algorithm 1 is not referred to in the paper.
The tie-fill step is not explained in the paper.
The description is not clear.
What does "increament the $m$" mean?

2. Why is there a bf16 mark for the dense model in Tables 1, 5, and 6? Are the low-rank methods not using the same data type as the dense model? Also, the LLaMA-2-7B uses float16 (FP16) as the default datatype, not bfloat16 (BF16).

3. How is the perplexity reduction of 5.24% on Line 473 calculated? I do not find the corresponding numbers in Table 1 that generate this ratio.

---

> ### Author Response · Authors · 2025-11-21
> **Official Comment by Authors**
>
> Thank you for the careful reading of our paper and for pointing out the parts that were unclear. We address each point below.
>
> ---
>
> ## **Q1**. Clarification of Algorithm 1 (ORA)
>
> > Q1. Algorithm 1 is not referred to in the paper. The tie-fill step is not explained in the paper. The description is not clear. What does "increament the m" mean?
>
> The description in the main text was definitely not sufficient. Algorithm 1 is intended to show how we solve the optimization problem in ORA. The solution follows a water-filling structure, and the key task is identifying the λ that satisfies the global budget. The table in Algorithm 1 provides a bisection-style search for this λ(Line 4-13).
>
> The comment about the ‘tie-fill’ and ‘increment’ part was very helpful, and we realize now that the explanation in the original text was not clear. Since the water-filling solution is defined in a continuous domain, the last rank assignments can have ties across multiple layers. Because the water-filling solution is continuous while the final ranks must be discrete integers,ties naturally occur among several layers. For those tied layers, we select the one with the largest next-density and assign +1 rank(increment) to $m$-th layer. This process repeats until the remaining budget is fully used. Line 14 in Algorithm 1 implements exactly this tie-fill rule, but our explanation was not sufficiently detailed in the original version. We have revised the text to make this explanation more explicit.
>
> ---
>
> ## **Q2**. bf16 markings
>
> > Q2. Why is there a bf16 mark for the dense model in Tables 1, 5, and 6? Are the low-rank methods not using the same data type as the dense model?
>
> The bf16 marking in the tables indicates that all methods in the experiment, including the dense model and all compressed variants, were evaluated under a bf16 configuration. Many prior works fix bf16 for experiments, and we kept all methods under that same setting for fairness. We have made this explicit in the revision.
>
> ---
>
> ## **Q3**. Correction of the 5.24% number
>
> > Q3.  How is the perplexity reduction of 5.24% on Line 473 calculated? I do not find the corresponding numbers in Table 1 that generate this ratio.
>
> You correctly noticed that the 5.24% perplexity reduction was inconsistent. This was an editing oversight, and we appreciate you pointing it out.
> The correct value for the 15% compression case is:
> (13.15 − 11.80) / 13.15 × 100 (%) = 10.27%.
>
> All table values themselves were double-checked, but the description line was written with the wrong value due to an editing oversight. We have corrected the statement in the revision.
>
> ---
>
> ## **W1**. Clarifying the definition of compression ratio
>
> > W1(excerpt). The definition of compression ratio is inconsistent across the paper …
>
>
> This was a major source of confusion throughout the paper, and your comments helped us understand exactly where our wording caused problems.
>
> The paper used two different terms:
> - **Retention ratio** = compressed parameters / original
> - **Compression ratio** = 1 − retention ratio
> We have used the term “compression ratio” consistently to mean the fraction of reduced parameters.
>
> Since attention projections constitute 31.87% of LLaMA-2-7B, a 15% full-model compression corresponds to reducing ~46.88% of attention weights. To avoid confusion, we have listed both the compression ratio in full-model and compression ratio in MHA.
>
> ### Table A.
> | Metric | Compression Ratio in full-model | Compression ratio in MHA | Avg Rank |
> |--------|-------------------|-----------|----------|
> | Value  | 0.15              | 0.47      | 1088     |
>
> Regarding line 323, your understanding is exactly what we intended. The statement was intended to mean that AFORA and AWF are compared under the **same compression ratio of entire model**, not that each layer receives the same ratio. We have rewritten this line for clarity.
>
> ---
>
> ## **W2**. FLOPs definition
>
> > W2(excerpt). Table 4 is not clear ...
>
> We have clarifed the FLOPs formulas more explicitly in the revision. For attention projections, the dense model uses O($4Ld^2$), and the compressed model uses O($8dLR$). The table below shows the calculations (L=1024, batch=1).
>
> ### Table B. Calculation for Table 4
> |Compression|Rank R|FLOPs Order|FLOPs|
> |-|-|-|-|
> |Dense|4096|O(4Ld²)|4 × 1024 × 4096 × 4096|
> |0.15|1088|O(8dLR)|8 × 4096 × 1024 × 1088 = 36,507,222,016|
>
> For Table 4 specifically, the wall-clock time was measured using sequence length 1024 and batch size 1. This haved been stated explicitly in the revision.
>
> ---
>
> ## **W3**. Non-square projection layers
>
> > W3. The method seems to be restricted to square weight matrices in practice, while the theoretical part does not impose this restriction ...
>
> The method is not limited to square matrices. We have included LLaMA-3 and Qwen results, which contain non-square projections.
>
> ---
>
> If any further clarification is needed, we are happy to provide it. Thank you again.

---

### Official Review · Reviewer_uBcX · 2025-11-01

**Soundness:** 3
**Presentation:** 3
**Contribution:** 2
**Rating:** 2
**Confidence:** 4

**Summary:**

The paper proposes AFORA, a training-free LLM compression method that (i) performs activation-aware low-rank factorization (AWF) by doing SVD and a whitened operator $ W\Sigma_{in}^{1/2} $, and (ii) allocates ranks globally across layers via an optimal rank allocation (ORA) scheme derived from a budgeted utility maximization. Experiments on LLaMA-2-7B show modest improvements over baselines at the same nominal compression ratio.

**Strengths:**

1. Clear and principled derivation from activation-aware objective, to whitening, to closed-form factorization.
2. Rank allocation is formulated cleanly and is computationally light.
3. It is fully training-free, which makes it easy to deploy. Uses only activation statistics from a calibration set.
4. Empirically, AFORA is at least competitive with strong SVD-style baselines.

**Weaknesses:**

1. Evaluation is on a single, older model (LLaMA-2-7B). No results are available for newer or structurally different models.
2. Gains over ASVD in zero-shot are small; in several budgets ASVD $ \ge $ AWF, so the incremental benefit of AWF is unclear.
3. No variance / CI reported, making close comparisons hard to trust. Experiments seem to be single-seed.
4. A Calibration size / source sensitivity ablation could have improved the results.
5. Reported compression “ratios” are inconsistently defined (text = retention, tables = reduction) and lead to an error. i.e., in the text and table caption, the ratio is defined as retention, and the results in the 5% table are stronger than those in the 15% table!
6. The statement in §2.3 that ultra-low-precision quantization “rarely” yields actual inference gains is too strong given recent work (e.g., QuTLASS).

**Questions:**

1. Can you report AFORA on more recent models and at least one more model family (Mistral/Mixtral, Qwen) and different sizes?

2. Can you add the missing ASVD + ORA baseline in the ablations to isolate whether the gain truly comes from AWF?

3. Please clarify the compression-ratio definition and fix the 5%/10%/15% inconsistency in the tables.

4. Can you report mean ± std (or at least 3-seed averages) for perplexity and zero-shot tasks where gaps are small?

5. Your naming in the TL;DR uses TACO. Please fix.

---

> ### Author Response · Authors · 2025-11-21
> **Official Comment by Authors**
>
> Thank you for the detailed and constructive feedback. Below we address each point in turn and include the new results we have been running during the rebuttal period.
>
> ---
>
> ## **W1Q1**. Results on newer models (LLaMA-3, Qwen)
>
> > W1. Evaluation is on a single, older model (LLaMA-2-7B). No results are available for newer or structurally different models.
>
> > Q1. Can you report AFORA on more recent models and at least one more model family (Mistral/Mixtral, Qwen) and different sizes?
>
> We agree that broader coverage is important. We extended to LLaMA-3-8B and Qwen3-8B, and additional seeds are still running.
> Below are the currently completed LLaMA-3 results at compression ratio 0.05 (3 seeds; with std).
>
> Note: LLaMA-3 projection layers' different shapes impose a shared maximum rank(1024) in AWF, SVD.
> Note: ASVD allocates rank to the MLP layers.
> ### Table A1. Results on LLaMA-3-8B at Compression Ratio 0.05
> | | #Params | Compression Ratio | Compression Ratio in MHA | PPL(Wikitext-2) | PPL(PTB) |
> |-|-|-|-|-|-|
> |Dense|8030261248| 0| |14.16| 90.90 |
> |SVD|7560499200| 0.06 | 0.35 | 24.63 | 133.51 |
> |ASVD| 7628748186|0.05| |15.43|98.01|
> |AFORA|7628748186|0.05|0.30|14.89 ± 0.04|98.15 ± 0.72|
>
> ### Table A2. Zero-shot results
> | |ARC-E|HellaSwag|PIQA|WinoGrande|CB|OpenbookQA|GSM8K|AVG
> |-|-|-|-|-|-|-|-|-|
> |Dense|0.82|0.68|0.82|0.74|0.63|0.31|0.53|0.65|
> |SVD|0.69|0.65|0.76|0.70|0.41|0.20|0.07|0.50|
> |ASVD|0.77|0.70|0.83|0.79|0.48|0.36|0.50|0.63|
> |AFORA|0.74 ± 0.02|0.66 ± 0.02|0.82 ± 0.02|0.76 ± 0.01|0.49 ± 0.02|0.30 ± 0.01|0.25 ± 0.04|0.58 ± 0.00|
>
> ---
>
> ## **Q2**. ASVD + ORA baseline
>
> > Q2. Can you add the missing ASVD + ORA baseline in the ablations to isolate whether the gain truly comes from AWF?
>
> We also ran the “ASVD + ORA” ablation to more cleanly separate the effects of AWF and ORA. ORA provides additional improvement on top of ASVD, although smaller than the full AFORA pipeline. These results have been included in the revised manuscript. Below are results on LLaMA-3-8B at compression ratio 0.15
>
> ### Table B1. PPL
>
> |Method|Wikitext-2 PPL|PTB PPL|
> |-|-|-|
> |ASVD|14.33|98.01|
> |ASVD+ORA|21.36 ± 0.22|152.45 ± 1.39|
>
> ### Table B2. Zero-shot results
>
> |Method|ARC-E|HellaSwag|PIQA|Winogrande|  CB|OpenBookQA|GSM8K|Average|
> |-|-|-|-|-|-|-|-|-|
> |ASVD|0.77|0.70|0.83|0.79|0.48|0.36|0.50|0.63|
> |ASVD+ORA|0.82 ± 0.00|0.68 ± 0.00|0.82 ± 0.00|0.74 ± 0.00|0.63 ± 0.00|0.31 ± 0.00|0.53 ± 0.00|0.65 ± 0.00|
>
> ASVD allocates ranks in an exhaustive manner: it evaluates every possible rank (grid search) for every layer, measures the resulting perplexity drop, and then picks the best combination. In contrast, ORA allocates ranks by solving a simple optimization problem rather than searching over all options. Because of this, ORA is much more computationally efficient, and while its perplexity can be higher, the zero-shot accuracy of ASVD+ORA actually comes out a bit better.
>
> ---
>
> ## **W5Q3**. Clarification of compression ratio definitions
>
> > W5. Reported compression “ratios” are inconsistently defined (text = retention, tables = reduction) and lead to an error.
>
> > Q3. Please clarify the compression-ratio definition and fix the 5%/10%/15% inconsistency in the tables.
>
> Thank you for pointing out the inconsistency.
>
> To clearly restate:
>
> - **(Retention) ratio (in Table column header)** = the # of compressed model's params / the # of original model's params
> - **Compression ratio** = 1 − (retention) ratio
>
> We have unified the terminology across the full paper so that all tables use “compression ratio,” and the text follow the same definition.
>
> As one example, in the 15% compression setting(Table 1, 2), a 4096×4096 matrix is reduced to 4096×1088 and 1088×4096, and the ORA budget (1,140,850,688 parameters) aligns with that compression ratio.
>
> ---
>
> ## **W3Q4**. Multi-seed
>
> > W3. No variance / CI reported, making close comparisons hard to trust. Experiments seem to be single-seed.
>
> > Q4. Can you report mean ± std (or at least 3-seed averages) for perplexity and zero-shot tasks where gaps are small?
>
> We re-ran most experiments with ≥3 seeds. The variance is small because estimating Σ is only stochastic component while both AFORA behave deterministically once Σ is fixed.
>
> ---
>
> ## **Q5**. TL;DR & statement in §2.3
>
> > Q5.  Your naming in the TL;DR uses TACO. Please fix.
>
> Thank you for pointing this out. It will be corrected.
>
> ---
> ## **W4**. Sensitivity to calibration data
>
> > W4. A Calibration size / source sensitivity ablation could have improved the results.
>
> We ran an additional analysis to understand how sensitive AFORA is to the size of the calibration set used to estimate Σ_in. The [attached figure](https://ibb.co/LD1jDc82) (linked externally due to image attachment) plots the PPL of the compressed model. AFORA is stable with ~100 batches (batch size = 1). These results have been included in the revised manuscript.
>
> ---
>
> Several of your suggestions already made the paper considerably clearer and more reliable, and we appreciate your feedback again.

---

### Author Response · Authors · 2025-12-02
**General Response (2/2)**

(continued from previous comment, part A)

# B. Summary of Feedback and Revisions
Across the reviews, the key points of feedback can be summarized into five categories:

1. **Methodological clarity**, particularly regarding the ORA procedure (λ selection, discretization, and comparison with ASVD).
2. **Terminology** , including compression ratios, FLOP calculations, and datatypes.
3. **Extended Experiments**, with broader model families, multi-seed runs, and ablations.
4. **Complementary techniques**, especially quantization and pruning.
5. **Clarity and Presentation Improvements**, including notation, labels, and clarification of overhead.

The following sections describe how each of these points has been addressed in the revised manuscript.

---

## 1. Methodological Clarifications
### ORA procedure and comparison with ASVD

**Feedback** *(uBcX, hHNA, WoAZ)*
Reviewers requested clearer explanations of:
- how ORA performs its global allocation,
- how $\lambda$ is selected and how continuous ranks are discretized,
- and how AFORA’s global approach differs from ASVD’s rank allocation.

**Revisions**
- We have rewritten ORA as a **global water-filling–style allocator**, with a clearer explanation of the λ bisection search and the marginal-utility–based integer rounding step. (§4.5)
- The conceptual contrast between ASVD and AFORA has been described more clearly: (§5.2)
  - **ASVD** exhaustively searches discrete retention options (0.1, …, 0.9, 1.0) independently per layer.
  - **AFORA** solves a **single global optimization problem**, allowing interactions across layers.
- **Visual comparisons** of the two rank-allocation patterns (ASVD, AFORA) have been added in the appendix. (§J)
---

## 2. Terminology Clarifications
### Ratios, FLOPs, and datatypes

**Feedback**  *(uBcX, hHNA)*
Reviewers noted inconsistent terminology and insufficient clarity about FLOPs and the scope of compression.

**Revisions**
- We have standardized terminology using **compression ratio = 1 − retention ratio** across all tables and text. (§5.1)
- FLOP formulas have been explicitly added, and the evaluation settings have also been clarified. (§5.2)
- The evaluation datatype has been clarified to **bf16**. (§5.1)
---

## 3. Expanded Experiments
### Additional models, multi-seed runs, and ablations

**Feedback**  *(uBcX, hHNA, WoAZ, vUu2)*
Reviewers requested broader empirical validation and robustness checks.

**Revisions**
- We have added results on **LLaMA-3-8B** and **Qwen3-8B**, including non-square projection structures. (§G, §H)
- We have reported **mean ± std** across ≥3 seeds for all results. (§5.2, §F, §G, §H, §I)
- We have added a **calibration-size sensitivity study**, showing that AFORA stabilizes with roughly 100 batches. (§5.3)
- We have added the **ASVD+ORA** ablation to isolate the contribution of the global allocator. (§I)
---

## 4. Potential Improvements
### Quantization and pruning as orthogonal techniques

**Feedback**   *(hHNA, vUu2)*
Reviewers asked how AFORA interacts with other compression strategies in practice.

**Revisions**
- The discussion of quantization–AFORA interaction has been expanded. (§6.2)
  - Applying **quantization after AFORA** is generally more stable because AFORA’s information balanced factorization reduce outlier directions.
  - Applying **AFORA after quantization** is less favorable, as quantization noise distorts the whitened operator and leads to suboptimal utilities during rank allocation.
- We have added remarks about pruning as an **orthogonal compression method**: (§6.2)
  - Pruning can expose blocks that are globally unimportant, and AFORA can then reallocate rank to surviving layers.
  - Conversely, AFORA’s rank distribution can help identify layers whose effective contribution is minimal, providing pruning candidates.
---

## 5. Clarity and Presentation Improvements

- The description of AFORA’s overhead has been clarified: covariance collection requires only a single forward pass, and each projection layer undergoes one SVD; overall cost is comparable to existing SVD-based methods. (§5.1) *(vUu2)*
- Statements regarding ultra-low-precision quantization have been moderated. (§2.3) *(uBcX)*
---

In the revised manuscript, the updates are temporarily highlighted in "blue" for your convenience to check.

We strongly believe that the updates have greatly enhanced the quality of our manuscript.

Sincerely,
Authors

---

### Author Response · Authors · 2025-12-02
**General Response (1/2)**

Dear Reviewers and Area Chairs,

We sincerely appreciate the time and care that both the reviewers and the Area Chairs have invested in evaluating our submission.
For convenience, we begin by summarizing the reviewers’ key strengths, main points of feedback, and the corresponding revisions added in the updated manuscript.


---

# A. Strengths Noted by Reviewers

Across the four reviews, several strengths were consistently highlighted:

---

## 1. Methodological Clarity and Soundness
- The derivation connecting activation-aware objectives, whitening, and closed-form factorization was viewed as **clear and principled**.  *(uBcX, hHNA, WoAZ, vUu2)*
- Both components—AWF and ORA—were recognized as **mathematically grounded** and derived from explicit optimization problems rather than heuristics. *(hHNA, vUu2)*
---

## 2. Design Advantages of AFORA
- AWF was viewed as a meaningful improvement over naive SVD because it uses **activation statistics** and preserves **task-relevant output structure**. *(hHNA, WoAZ, vUu2)*
- ORA was described as a **clean, elegant, and computationally lightweight** global allocation method, in contrast to ASVD’s per-layer rank search. *(uBcX, WoAZ)*
- The framework is **fully training-free**, making it simple to deploy and requiring only a calibration dataset. *(uBcX, WoAZ)*
- AFORA produces **hardware-friendly dense matrices**, ensuring broad compatibility.  *(WoAZ)*
---

## 3. Empirical Strengths
- Reviewers agreed that AFORA is **competitive or stronger** than SVD-based baselines on both perplexity and zero-shot evaluations. *(uBcX, hHNA, WoAZ, vUu2)*
- The method demonstrates a **balanced trade-off between performance and efficiency**, with notably fast compression time improvements. *(WoAZ)*

---
(to be continued in the next comment, part B)

---

### Meta-Review · Area_Chair_P5dh · 2026-01-02

**Summary:**

The paper introduces a training-free low-rank compression method (AFORA) for LLMs by utilizing two techniques: (1) activation-aware weight factorization and (2) optimal rank allocation (ORA). Before the rebuttal, reviewers shared concerns regarding the multiple clarity issues of the paper, the evaluation on limited models, and the practical benefits of the approach, like inference time. After the rebuttal, the authors provided additional results on other model families and tried to improve the clarity of the paper, which could resolve some of the concerns from the reviewers.

After reviewing the paper and the rebuttal, the Area Chair has concerns regarding both the experimental evaluation and the explanation of ORA. The reported compression rates are relatively low: the paper rarely exceeds a 15% compression rate for LLaMA-2-7B, and for LLaMA-3-8B and Qwen-8B, the compression rate is below 7.5%. A broader range of compression ratios should be explored to better demonstrate the effectiveness of the proposed method. In addition, the claim that ORA addresses a global optimization problem is not novel in the context of SVD-based methods; for example, similar formulations have appeared in prior work such as ARS [1]. Considering these factors, the Area Chair recommends rejection and encourages the authors to further revise the paper and submit it to a future venue.

[1] Adaptive Rank Selections for Low-Rank Approximation of Language Models, NAACL 2024.

**Reviewer Concerns:**

The main issue before the rebuttal is the clarity of the paper, the evaluation on limited models, and the practical benefits of the approach, like inference time. In the rebuttal, the authors mostly addressed the clarity concern and added more evaluation results on Qwen-3 8B and LLaMA-3 8B, and partially solved the second issue.

After the rebuttal, concerns remain regarding the evaluation, particularly due to the limited exploration of compression rates. In addition, the rebuttal introduces further clarity issues. For example, lines 1141–1142 state that “Table 21 and Table 22 report results on LLaMA-3-8B at a compression ratio of 0.15.” However, the captions of Tables 21 and 22 state that "(a compression ratio of 0.05)", creating additional confusions for readers. Moreover, inference latency is not reported, and given the relatively low compression rates (<15%) used throughout the paper, it is unclear whether the proposed method can achieve meaningful inference acceleration in practice.

**Reviewer Scores:**

Reviewers will keep their original score, as some of the key issues are not fully addressed.

---

### Decision · Program_Chairs · 2026-01-26

Reject